# HCMV-encoded US7 and US8 act as antagonists of innate immunity by distinctively targeting TLR-signaling pathways

Areum Park[1], Eun A. Ra[1], Taeyun A. Lee[1], Hyun jin Choi[1], Eunhye Lee[1], Sujin Kang[1], Jun-Young Seo[2], Sungwook Lee[3] & Boyoun Park[1]*

The mechanisms by which many human cytomegalovirus (HCMV)-encoded proteins help the virus to evade immune surveillance remain poorly understood. In particular, it is unknown whether HCMV proteins arrest Toll-like receptor (TLR) signaling pathways required for antiviral defense. Here, we report that US7 and US8 as key suppressors that bind both TLR3 and TLR4, facilitating their destabilization by distinct mechanisms. US7 exploits the ER-associated degradation components Derlin-1 and Sec61, promoting ubiquitination of TLR3 and TLR4. US8 not only disrupts the TLR3-UNC93B1 association but also targets TLR4 to the lysosome, resulting in rapid degradation of the TLR. Accordingly, a mutant HCMV lacking the US7-US16 region has an impaired ability to hinder TLR3 and TLR4 activation, and the impairment is reversed by the introduction of US7 or US8. Our findings reveal an inhibitory effect of HCMV on TLR signaling, which contributes to persistent avoidance of the host antiviral response to achieve viral latency.

[1] Department of Systems Biology, College of Life Science and Biotechnology, Yonsei University, Seoul 03722, South Korea. [2] Severance Biomedical Science Institute, Brain Korea 21 PLUS Project for Medical Science, Yonsei University College of Medicine, Seoul 03722, South Korea. [3] Division of Tumor Immunology, Research Institute, National Cancer Center, Goyang 10408, South Korea. *email: bypark@yonsei.ac.kr

Human cytomegalovirus (HCMV), a member of the beta herpesvirus family, can establish lifelong latency after primary infection. HCMV has developed a variety of strategies to evade innate and adaptive immune responses through co-evolution with its host[1,2]. Although HCMV expresses a large number of membrane glycoproteins with unknown functions, the 9 kb US2-US11 region of the unique short (US) part of the HCMV genome encodes two miRNAs (US4 and US5) and a five of eight type I glycoproteins (US2, US3, US6, US10, and US11), which are known to inhibit the activation of innate or adaptive immune responses[3–7]. Recently, it was demonstrated that HCMV US9 blocks natural killer (NK) cell activation and interferon (IFN)-β production by targeting signaling mediated by major histocompatibility complex (MHC) class I-related chain A (MICA)*008 and by mitochondrial antiviral signaling protein (MAVS) and stimulator of interferon genes (STING), respectively[5,8]. However, the cellular targets and functions of US7 and US8 are yet unknown.

Type I IFN and pro-inflammatory cytokines induced by HCMV infection play important roles in activating innate immune responses that interfere with viral replication and intercellular transmission. HCMV has therefore developed mechanisms to block the innate antiviral response. Specifically, HCMV inhibits cytokine production by targeting signaling molecules or by accelerating the turnover of cytokine-encoding mRNAs[9–14]. In addition, HCMV blocks IFN production by disrupting multiple levels of the IFN signal-transduction pathway[15–19]. It is largely unclear, however, whether HCMV targets Toll-like receptors (TLRs), which are major cellular sensors for triggering type I IFNs and pro-inflammatory cytokines.

TLRs are pattern recognition receptors and part of the first line of defense against infective pathogens. Thus, the TLR family members specifically recognize microbial or viral products on the surface of cells or in endolysosomes and strongly trigger inflammatory responses to eliminate infections[20–22]. In particular, TLR3 detects double-stranded RNA (dsRNA), which is generated by both RNA and DNA viruses[23–25], and TLR4 recognizes Gram-negative bacterial lipopolysaccharide (LPS) as well as viral components such as envelope glycoproteins[26–29]. Upon binding its ligand, TLR3 induces the activation of IFN-regulatory factor 3 (IRF-3) and nuclear factor-kappa B (NF-κB) via the adapter molecule TIR-domain-containing adapter-inducing interferon-β (TRIF), which drives the production of type I IFNs and pro-inflammatory cytokines[30]. TLR4 signal transduction can occur via a myeloid differentiation factor 88 (MyD88)-dependent and a TRIF-dependent pathways.

Several viruses have evolved specific proteins that target TLRs to limit the release of IFN or pro-inflammatory cytokines, thus exploiting an attractive mechanism for disturbing the innate immune response. Hepatitis C virus protease NS3/4 A cleaves TRIF to block TLR3-mediated signaling[31]. Vaccinia virus-encoded A46R associates with TLR adaptors and thereby disrupts the activation of NF-κB and IRF3[32]. Kaposi's sarcoma-associated herpesvirus (KSHV) and hepatitis B virus also inhibit TLR2, TLR4, and TLR9 expression, leading to a reduction of pro-inflammatory cytokines[33–35]. Herpes simplex virus immediate-early ICP0 has an E3 ligase activity that promotes the degradation of the TLR2 adaptor protein and the inhibition of the NF-κB signaling[36]. The microRNA (miRNA) UL112-3p produced by HCMV represses the expression of TLR2;[37] however, it is yet unknown whether any HCMV-encoded proteins directly target the TLRs.

In this study, we demonstrate that the HCMV-encoded US7 and US8 glycoproteins function as suppressors of the innate immune response by targeting TLR3 and TLR4. US7 promotes ubiquitin-dependent and proteasome-dependent degradation of TLR3 and TLR4, while US8 facilitates the disruption of the TLR3-UNC93B1 interaction and the endolysosomal translocation of TLR4, leading to TLR3 and TLR4 instability and subsequent blockade of antiviral responses. Consistent with those observations, HCMV infection suppresses TLR3 and TLR4 gene expression and IFN production in vivo. Our findings propose a crucial mechanism by which HCMV glycoproteins US7 and US8 directly target TLR signaling and thus contribute to the evasion of the host antiviral immune response.

## Results

**US7 and US8 target TLR signaling pathways.** The double-stranded (ds)DNA genome of HCMV is composed of unique long (UL) and unique short (US) regions, which are flanked on one end by terminal repeat sequences (TR$_L$/TR$_S$) and on the other end by internal repeats (IR$_L$/IR$_S$) (Fig. 1a). The genes in the US2–US11 region target essential stages of antigen presentation to CD8$^+$ T cells (US2, US3, US4, US6, and US11), MAVS-mediated or STING-mediated IFN-β production (US9), NK cell activation (US9 and US10), and NF-κB-mediated cytokine production (US5); however, the targets of US7 and US8 have not yet been identified (Fig. 1a). To identify cellular targets of US7 and US8, we stimulated HCMV-permissive human foreskin fibroblast (HFF) cells expressing hemagglutinin (HA)-tagged US7 (HA-US7) or US8 (HA-US8) with dsDNA and the changes in gene expression were monitored using microarrays. As expected, the dsDNA-stimulated HFF cells showed a robust increase in the expression of various immune-related genes, such as IFN-related genes, pro-inflammatory cytokines, and chemokines. Notably, the dsDNA-induced expression of immune-related genes in HFF cells expressing US7 or US8 was significantly reduced compared with that in control cells (Fig. 1b). Among the genes with the greatest decrease in expression mediated by US7 or US8 were *ifnb*, *tnfsf10*, *ccl8*, *cxcl10*, *cxcl11*, *ifit3*, and *isg15* (Fig. 1c). To confirm those results obtained using microarrays, we performed quantitative real-time PCR (qPCR) analysis using dsDNA-stimulated HFF cells that stably expressed empty vector, HA-US7, or HA-US8. US7 or US8 expression consistently resulted in significantly lower expression of *ifnb*, *tnfsf10*, *ccl8*, *cxcl10*, *cxcl11*, *ifit3*, and *isg15* (Fig. 1d). These results suggest that HCMV glycoproteins US7 and US8 target the innate immune response.

US7 and US8 each possess a putative signal sequence at the end of their N-terminal region, we thus further examined their subcellular localization by immuno-fluorescence assay (IFA). US7 predominantly localized in the endoplasmic reticulum (ER), while US8 appeared in the Golgi or lysosome, and it clearly accumulated in lysosomes in the presence of chloroquine, an inhibitor of lysosomal acidification (Supplementary Fig. 1a-c). Because IFN production is highly linked to the ER, ER-mitochondria junctions, and lysosomes, where STING/MAVS and TLRs are located[38], we hypothesized that US7 or US8 may target immune sensors in the ER or lysosome to prevent host antiviral responses. To test that, we first examined the effects of US7 and US8 on *ifnb* expression in cells stimulated by STING or MAVS overexpression, which activates the STING or MAVS signaling cascade; however, there was no difference in *ifnb* expression among cells expressing empty vector, US7-GFP, and US8-GFP (Supplementary Fig. 2a). To further assess whether US7 or US8 affect TLR-mediated signaling, we examined their effects on cytokine production in cells stimulated with Pam3CSK4, synthetic dsRNA (poly(I:C)), LPS, Imiquimod, or CpG-DNA, which robustly activate the TLR2, TLR3, TLR4, TLR7, and TLR9 signaling cascades, respectively. HFF cells expressing US7 or US8 showed impaired TLR-mediated IL-6 production after stimulation with the TLR-activating agents compared with cells

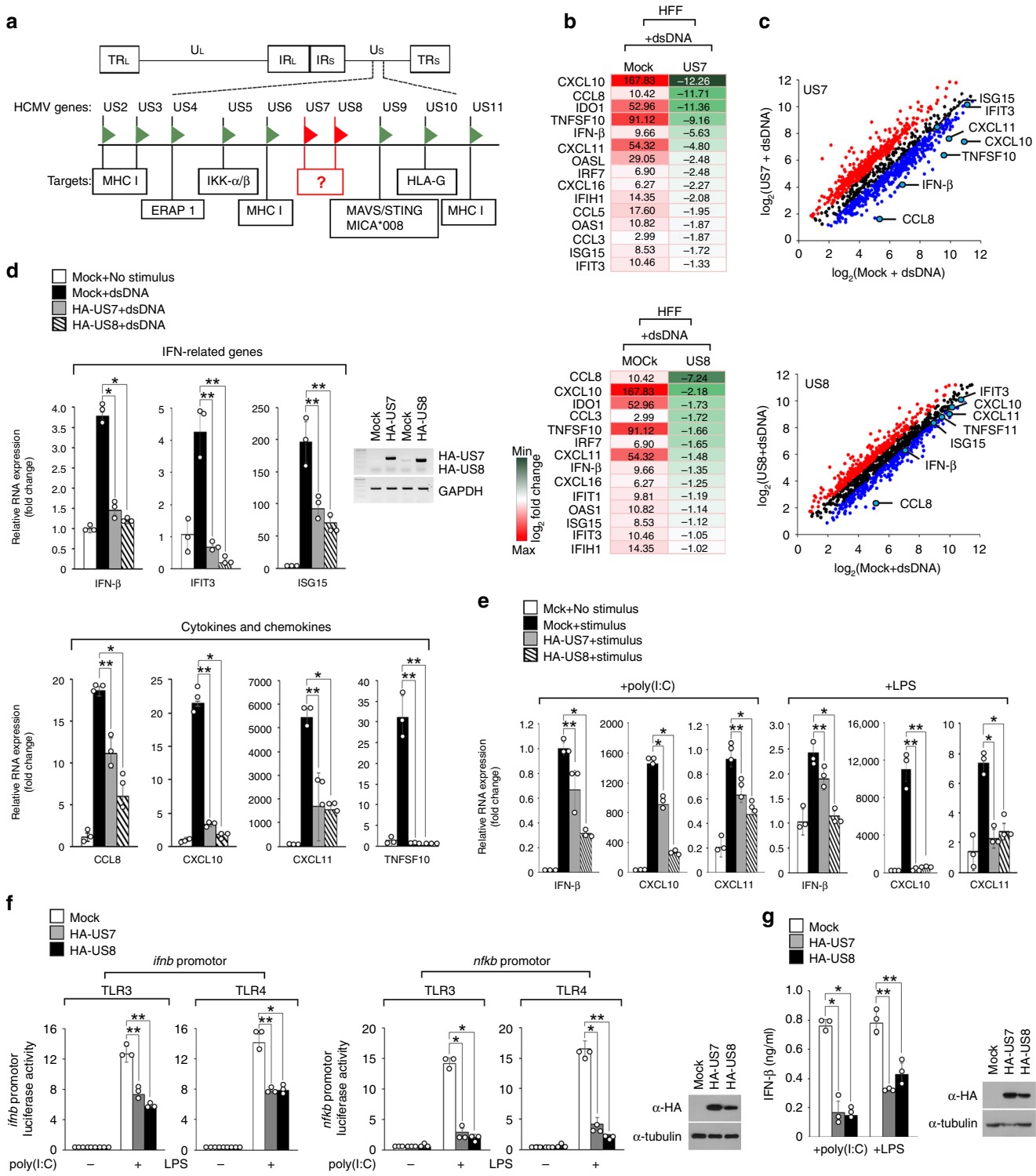

expressing empty vector (Supplementary Fig. 2b). Particularly, since TLR3 and TLR4 play an important role in the stimulation of IFN-β production and subsequent activation of protective innate immunity against viral infection[20–22], we focused on determining whether TLR3 or TLR4 is responsible for activating IFN production through the TRIF pathway. To assess whether US7 or US8 affects TLR3-mediated or TLR4-mediated signaling, we examined the effects of US7 and US8 on type I IFN and cytokine production in cells stimulated with synthetic dsRNA (poly(I:C))

or LPS, which robustly activate the TLR3 and TLR4 signaling cascades, respectively. HFF cells expressing US7 or US8 showed impaired TLR3-mediated or TLR4-mediated transcription of *ifnb*, *cxcl10*, and *cxcl11* genes compared with cells expressing empty vector (Fig. 1e) To confirm the qPCR results, we measured the effects of US7 and US8 on TLR3-mediated and TLR4-mediated *nfkb* and *ifnb* promoter activity in HEK293T cells, which have a higher transfection efficiency than HFF cells. Consistently, luciferase reporter assays showed that the *nfkb* and *ifnb* promoter

**Fig. 1** HCMV US7 and US8 target TLR3-mediated and TLR4-mediated antiviral responses. **a** Schematic representation of the HCMV genome and the US2-US11 region capable of targeting various cellular immune molecules. **b** Heat map showing expression of cellular targets of US7 and US8 in HFF cells expressing US7 or US8 after stimulation by dsDNA (Supplementary Data 1). **c** Change in cellular mRNA expression caused by US7 or US8 versus mRNA expression in empty vector-expressing HFF cells stimulated by dsDNA. Scatter plots of US7- or US8-upregulated ( > 1.5-fold change, red dots) or -downregulated genes ( < 1.5-fold change, blue dots) in dsDNA-stimulated HFF cells. **d** US7 and US8 inhibit DNA-induced innate antiviral response. HFF cells expressing empty vector, HA-US7, or HA-US8 were transfected with 500 ng ml$^{-1}$ dsDNA for 12 h. The mRNA expression of the indicated genes was analyzed by qPCR or RT-PCR. *$P < 0.001$, **$P < 0.05$ (Student's $t$-test). **e** US7 and US8 inhibit antiviral gene expression mediated by TLR3 and TLR4. TLR3-Myc- or TLR4-Myc-expressing HFF cells were transduced with empty vector, HA-US7, or HA-US8 and were then stimulated by 10 μg ml$^{-1}$ poly(I:C) or 5 μg ml$^{-1}$ LPS for 12 h. The indicated gene expression was measured by qPCR. *$P < 0.001$, **$P < 0.05$ (Student's $t$-test) **f** US7 and US8 suppress *ifnb* and *nfkb* promoter activity. Luciferase assays of *ifnb* and *nfkb* promoter activity in TLR3- or TLR4/MD2-expressing HEK293T cells transfected with empty vector, HA-US7, or HA-US8 and incubated with 5 μg ml$^{-1}$ LPS or 10 μg ml$^{-1}$ poly(I:C) for 12 h. The protein over-expression of HA-US7 or HA-US8 was analyzed by immunoblot analysis with anti-HA antibody. *$P < 0.001$, **$P < 0.05$ (Student's $t$-test). **g** US7 and US8 block IFN-β production mediated by TLR3 and TLR4. THP-1 cells expressing empty vector, US7, or US8 were stimulated by 100 μg ml$^{-1}$ poly(I:C) or 1 μg ml$^{-1}$ LPS for 24 h. IFN-β secretion levels or HA-US7/US8 overexpression levels were analyzed by ELISA or immunoblot analysis, respectively. *$P < 0.001$, **$P < 0.05$ (Student's $t$-test). Data are representative of three independent experiments and are presented as the mean ± s.d. in **d–g**. Source data are provided as a Source Data file

---

activity evoked by dsRNA or LPS in HEK293T cells expressing US7 or US8 was significantly lower than that in control cells (Fig. 1f). We observed a significant difference in the mRNA expression of *cxcl10* in cells expressing US7 or US8, but not in cells expressing other US proteins including US14, and US15 (Supplementary Fig. 2c). To determine if the abilities of US7 and US8 to block TLR3 and TLR4 signaling are cell-type specific, we observed IFN-β protein secretion in US7-expressing or US8-expressing immune cells of the monocyte-macrophage lineage, THP-1. We found that TLR3-mediated and TLR4-mediated IFN-β secretion levels were reduced in cells that stably expressed US7 or US8 (Fig. 1g). Those results suggest that both US7 and US8 are suppressors of IFN production induced by TLR3 or TLR4.

**US7 and US8 facilitate the degradation of both TLR3 and TLR4.** To explore how US7 and US8 inhibit signaling mediated by TLR3 and TLR4, we examined the mRNA and protein levels of Myc-tagged TLR3 and TLR4 in cells expressing HA-US7 or HA-US8. Notably, both US7 and US8 decreased TLR3 and TLR4 protein levels, but not mRNA levels (Supplementary Fig. 3a and b). To confirm those results, we examined whether US7 or US8 downregulates endogenous TLR3 and TLR4 protein levels. To test that, we assessed the endogenous expression levels of TLR3 and TLR4 in HeLa cells expressing empty vector, US7, or US8 by immunoblot assay with anti-TLR3 and anti-TLR4 antibodies. Consistent with the results of the overexpression system, the endogenous TLR3 and TLR4 protein levels were reduced by US7 or US8 expression (Fig. 2a). Because newly synthesized TLR4 is present at the plasma membrane, where it recognizes molecular components on the surface of pathogens, we next examined the cell-surface levels of endogenous TLR4 by flow cytometry. Cells expressing HA-US7 or HA-US8 had reduced surface expression of TLR4 compared with control cells expressing other US protein, US3 (Fig. 2b and Supplementary Fig. 3c). We further confirmed our results by biochemical analysis of cells labeled with [$^{35}$S]-methionine and cysteine. In cells expressing HA-US7 or HA-US8, despite similar rates of TLR4 synthesis during the 1 h pulse, US7 and US8 degraded 57.7% and 65.6% of the labeled TLR4, respectively, after the 4 h chase (Fig. 2c). Moreover, an endoglycosidase H (Endo H)-resistant form of TLR4 was observed in control cells but was barely detectable in cells expressing US7 or US8 (Fig. 2c, asterisk). Those results suggest that HCMV US7 and US8 reduce the stability of TLR3 and TLR4, and particularly the surface expression of TLR4.

To investigate whether the proteasome or lysosome is involved in US7-mediated or US8-mediated degradation of TLR3 or TLR4, we monitored those TLR levels in HeLa cells that stably expressed US7 or US8 and Myc-tagged TLR3 (TLR3-Myc) or TLR4-Myc in

the presence of the MG132, a proteasome inhibitor, or chloroquine. In US7-expressing cells, MG132, but not chloroquine, restored the levels of TLR3 and TLR4, whereas TLR3 and TLR4 degradation was drastically attenuated by both MG132 and chloroquine in US8-expressing cells (Fig. 2d, e). Collectively, these results suggest that US7 destabilizes TLR3 and TLR4 through a proteasome-dependent pathway, whereas US8-mediated degradation of TLR3 and TLR4 proteins involves both the proteasome and lysosome.

**US7 degrades TLR3 and TLR4 by a ubiquitin/proteasome system.** To elucidate whether US7-induced TLR3 and TLR4 degradation involves direct interaction between US7 and both TLRs, we transfected HA-US7 into HeLa cells and then performed co-immunoprecipitation (IP) experiments with cells treated with MG132 to prevent US7-mediated degradation to obtain sufficient amounts of endogenous TLR3 or TLR4 proteins. US7 clearly bound to both endogenous TLR3 and TLR4, which was consistent with IFA data showing that US7 co-localizes with TLR3 and TLR4 (Fig. 3a, b).

Previous studies have proposed that US2 or US11 uses Sec61 or Derlin-1, essential ER-associated degradation components that facilitate the translocation of substrates such as MHC class I molecules from the ER to the cytosol, leading to ubiquitin-dependent destruction in the proteasome[39–41]. We thus determined whether Sec61 or Derlin-1 is involved in US7-mediated TLR3 and TLR4 degradation by assessing the physical interaction between HA-US7 and endogenous Sec61 and Derlin-1. Interestingly, the ability of US7 to bind both Sec61β and Derlin-1 was similar to that of US2 or US11 (Fig. 3c). To exclude tagging specificities, we assessed the binding of Sec61β to GFP and GFP-US7. Consistent with the results for HA-US7, Sec61β bound to GFP-US7, but not to GFP alone (Supplementary Fig. 3d). To further examine the involvement of Sec61β and Derlin-1 in US7-mediated endogenous TLR3 and TLR4 degradation, we used small hairpin RNAs (shRNAs) to deplete Sec61β or Derlin-1 in HeLa cells. Although the effects seemed to depend on the knockdown efficiency of the shRNAs, we found that US7-induced degradation of endogenous TLR3 and TLR4 was considerably restored by the depletion of Sec61β or Derlin-1 (Fig. 3d). Because US7-mediated TLR3 and TLR4 degradation depends on a proteasomal pathway, we next examined whether US7 promotes the ubiquitin-dependent breakdown of TLR3 or TLR4. US7-expressing cells displayed considerable accumulation of ubiquitin conjugates of TLR3 and TLR4 in the presence of MG132 (Fig. 3e). Collectively, these results suggest that US7 degrades TLR3 and TLR4 by a ubiquitin/proteasome system via its association with Sec61β or Derlin-1.

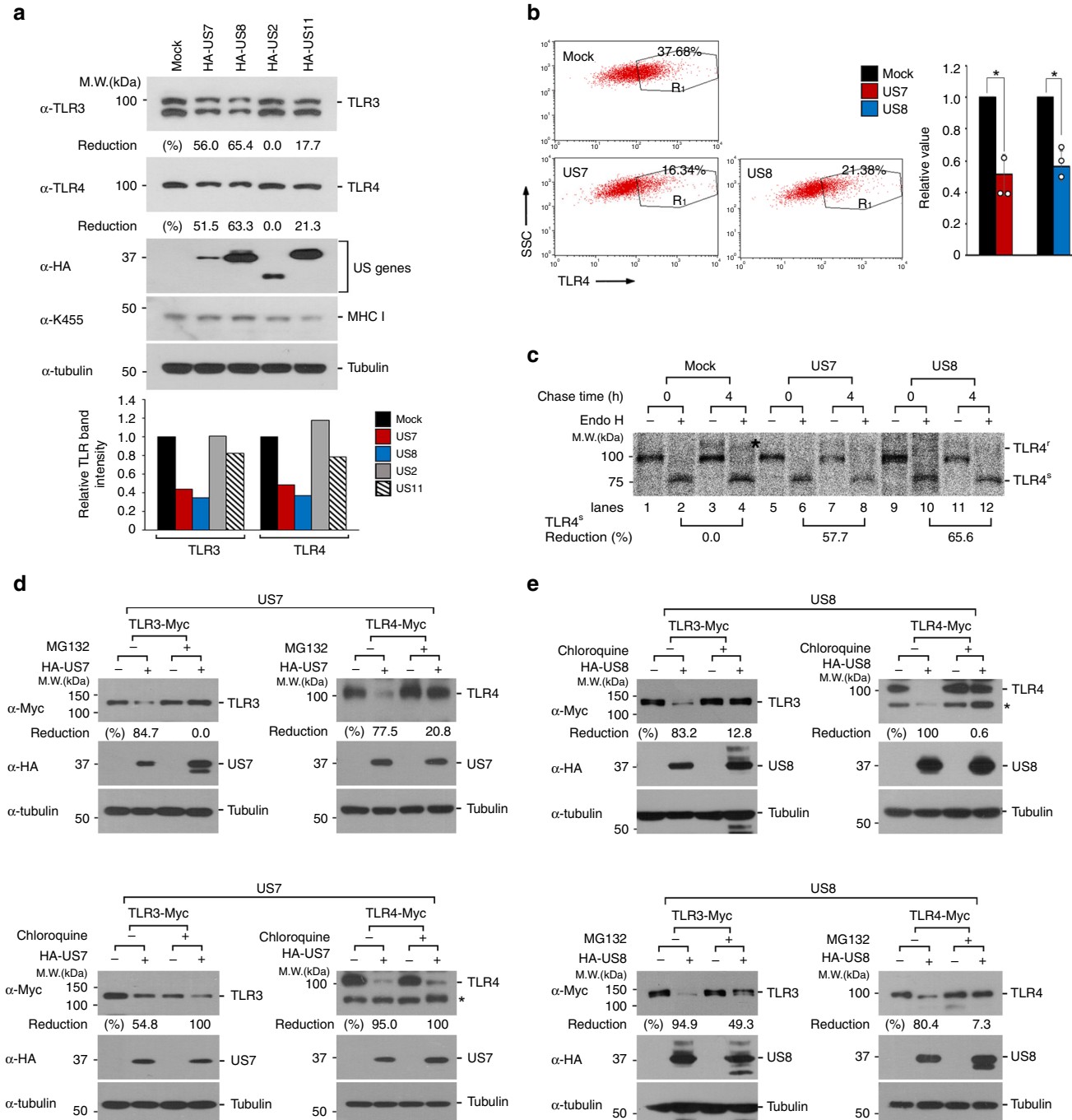

**Fig. 2** HCMV US7 and US8 destabilize TLR3 and TLR4. **a** US7 and US8 degrade TLR3 and TLR4. HeLa cells were transfected with empty vector, HA-US2, HA-US7, HA-US8, or HA-US11 and then the lysates were immunoblotted with anti-TLR3, anti-TLR4, anti-HA, anti-MHC class I molecules (K455), or anti-Tubulin antibody. The intensity of TLRs bands was quantified as comparing the relative abundance of TLR3 or TLR4 to Tubulin (bottom graphs). **b** US7 and US8 downregulate the cell-surface expression of TLR4. Endogenous TLR4 surface expression on U937 cells expressing US7 or US8 was assessed by FACS analysis. Right graph, validation of TLR4 cell surface expression levels in US7- or US8-expressing cells by FACS. *$P < 0.001$ (Student's $t$-test). **c** HeLa cells stably expressing TLR4-Myc were transfected with HA-US7 or HA-US8. Cells were labeled with [$^{35}$S] methionine/cysteine for 1 h and chased for 4 h. After endo-H treatment, lysates were immunoprecipitated with anti-Myc antibody. Asterisk denotes an endo-H resistant form of TLR4. **d**, **e** US7 and US8 destabilize TLR3 and TLR4 by a proteasome-dependent and/or lysosome-dependent pathway. HeLa cells expressing TLR3-Myc or TLR4-Myc were transfected with empty vector, HA-US7 **d**, or HA-US8 **e** and then treated with either 20 μM MG132 or 100 μM chloroquine for 4 h. Lysates were immunoblotted with the indicated antibodies. Asterisk indicates a non-specific bands. Data are representative of at least three independent experiments and are presented as the mean ± s.d. in **b**. Source data are provided as a Source Data file

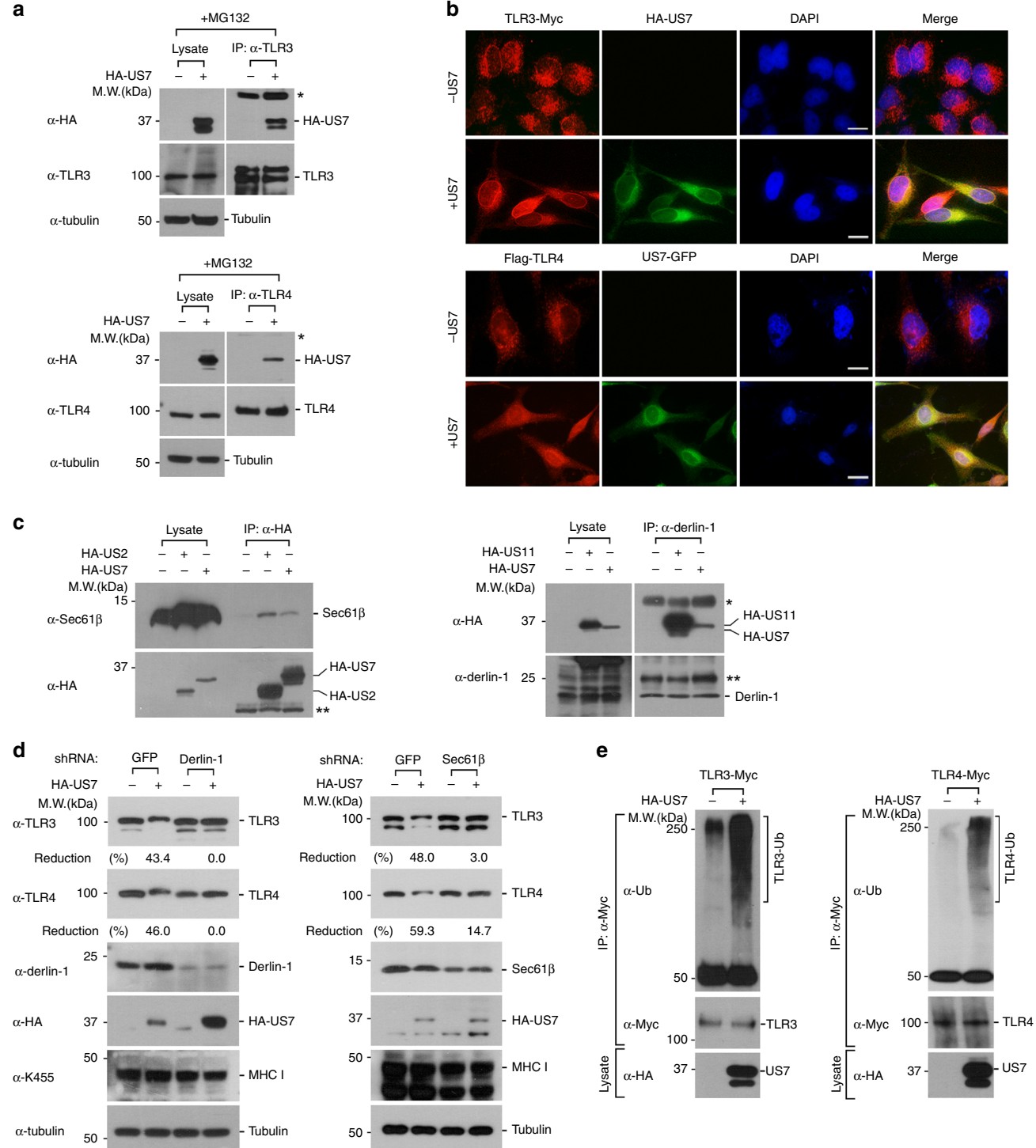

**Fig. 3** US7-mediated TLR3 or TLR4 degradation requires Derlin-1/Sec61 and ubiquitination. **a** US7 physically interacts with TLR3 and TLR4. HeLa cells were transfected with empty vector or HA-US7 and treated with 20 μM MG132 for 4 h. Lysates were immunoprecipitated with anti-TLR3 or anti-TLR4 antibody and then immunoblotted with anti-HA, anti-TLR3, anti-TLR4, or anti-Tubulin antibody. Asterisk denotes immunoglobulin (Ig) heavy chains. **b** US7 co-localizes with TLR3 or TLR4. HeLa cells expressing TLR3-Myc or Flag-TLR4/MD2-Myc were transfected with empty vector, US7-GFP or HA-US7 and then stained with anti-Myc, anti-Flag or anti-HA antibody. DAPI was used as a nuclear counterstain. Data are given as the average fluorescence intensity (FI) per cell in the selected fields. Scale bars, 10 μm. **c** US7 binds to both Sec61β and Derlin-1. Lysates from HA-US7-expressing HEK 293 T cells were immunoprecipitated with anti-HA and anti-Derlin-1 and immunoblotted with anti-Sec61β, anti-Derlin-1, or anti-HA antibody. HA-US2 or HA-US11 served as a control. *, Ig heavy chains; **, Ig light chains. **d** US7-mediated TLR3 and TLR4 degradation depends on Sec61β and Derlin-1 expression. HeLa cells stably expressing control shRNA, Sec61β, or Derlin-1 were transfected with HA-US7. Lysates were immunoblotted with the indicated antibodies. **e** US7 promotes TLR3 and TLR4 ubiquitination. Cells expressing TLR3-Myc or TLR4-Myc with or without HA-US7 were treated with 20 μM MG132 for 4 h and lysed. Cell extracts were immunoprecipitated with anti-Myc antibodies and then analyzed by immunoblotting with anti-Ubiquitin, anti-HA, anti-Tubulin and anti-Myc antibodies. Data are representative of at least three independent experiments. Source data are provided as a Source Data file

**US8 promotes TLR3 and TLR4 destabilization**. Because US8 is detectable in lysosomal compartments, and its ability to degrade TLR3 and TLR4 is impaired in the presence of chloroquine, we speculated that US8 directly targets TLR3 and TLR4 to the lysosome via a physical interaction with the TLRs. Indeed, US8 associated with both endogenous TLR3 and TLR4, which is concordant with the IFA data showing the colocalization of US8 with both tagged TLR proteins (Fig. 4a, b). TLR4 mainly localized in either the ER/Golgi or the plasma membrane in control cells, but it endocytosed to the early endosome in LPS-stimulated cells (Supplementary Fig. 4), which promotes the subsequent expression of IFNs and IFN-related genes[42]. Intriguingly, regardless of LPS stimulation, the TLR4 distribution included punctate structures in lysosomal compartments containing US8 (Fig. 4b and Supplementary Fig. 4). The TLR4 distribution displayed dim punctate structures in lysosomal compartments containing US8. Interestingly, those structures were exhibited more clearly in the presence of chloroquine, suggesting that US8 targets TLR4 to the lysosome for its degradation (Fig. 4b). Additionally, in accordance with previous findings showing that US8 degrades TLR4 in a proteasome-dependent manner (Fig. 2e), US8 also enabled TLR4 to be ubiquitinated (Fig. 4c), presumably because of the structural destabilization of TLR4 caused by the physical interaction with US8. Together, our results suggest that US8 promotes aberrant translocation of TLR4 into lysosomes and ubiquitination for lysosomal or proteasomal degradation.

Because TLR3 originally requires translocation to the endolysosomes, where it plays an important role in the activation of TLR3 signaling, we needed to understand how US8 promotes a decrease in TLR3 stability or activation in endolysosomes. To explore that, we focused on the effect of US8 on the regulation and function of UNC93B1, which enables TLR3 to be transported to endolysosomes and to stabilize[43–46]. Co-IP experiments revealed that US8 binds to UNC93B1, which is consistent with IFA data showing that UNC93B1 was targeted to US8-containing lysosomal compartments (Fig. 4d, e). Overexpression of US8 decreased the level of UNC93B1, and US8-mediated UNC93B1 destabilization was rescued by chloroquine (Fig. 4f). We also observed that US8 inhibits the TLR3-UNC93B1 interaction (Fig. 4g). Furthermore, TLR3 that was rendered incapable of binding to UNC93B1 by US8 was eventually ubiquitinated and degraded in a proteasome-dependent manner, which was similar to the pattern observed with TLR4 (Fig. 4h, compared with Fig. 2e). These results suggest that US8 facilitates TLR3 destabilization by blocking the binding affinity of UNC93B1 for TLR3, thus promoting the lysosomal degradation of UNC93B1. Taken together, our findings demonstrate that US8 disrupts the TLR3-UNC93B1 interaction and targets TLR4 into lysosomes, thereby promoting TLR3 and TLR4 destabilization through proteasome-dependent or lysosome-dependent degradation.

**The C terminus of US7 and US8 is involved in their function**. Because the C-terminal region of US7 and US8 faces the cytoplasmic side of the cellular membrane, where many immune regulatory proteins elicit their immune responses, we speculated that the C-terminal region might contribute to TLR3 or TLR4 degradation and thus block IFN or cytokine production. To test this, we constructed deletion mutants lacking the C-terminal domain of US7 and US8 (US7ΔCT and US8ΔCT; Fig. 5a) and measured the stability and fast-mobility of the mutant proteins in non-reducing conditions to assess whether they undergo proper folding. The protein stability and mobility of US7ΔCT and US8ΔCT were similar to those of wild-type US7 and US8, suggesting that the mutants maintained their proper folding

(Supplementary Fig. 5a, b). The intracellular distribution and the Sec61/Derlin-1 interaction of US7ΔCT were both similar to those of wild-type US7 (Supplementary Fig. 6a and b). Notably, US7ΔCT-expressing cells showed restored *cxcl10* and *ifnb* mRNA expression compared with wild-type US7-expressing cells (Fig. 5b). Consistent with the qPCR results, cells that expressed US7ΔCT exhibited normal luciferase activity at the *ifnb* promoter after stimulation with dsRNA or LPS (Supplementary Fig. 6c). Although the expression levels of the wild-type and mutant proteins were comparable, US7ΔCT was ineffective in causing TLR3 or TLR4 degradation and ubiquitination compared with wild-type US7 (Fig. 5c, d). Likewise, in contrast to wild-type US7, US7ΔCT resulted in normal cell-surface expression of TLR4, which was almost equal to that observed in control cells (Fig. 5e). These results suggest that the cytoplasmic region of US7 is required for degrading TLR3 and TLR4 proteins.

We further analyzed whether the cytoplasmic C-terminal domain of US8 is essential for blocking TLR3 or TLR4 activation. In contrast to full-length US8, US8ΔCT failed to localize to lysosomes (Supplementary Fig. 7a), suggesting that the cytoplasmic domain of US8 is involved in its subcellular translocation to the lysosome. Of note is that the C-terminal region of US8 contains two putative tyrosine-based motifs responsible for lysosomal targeting[47]. To determine if those tyrosine motifs are essential for the lysosomal targeting of US8, we generated a point mutation in the US8 cytoplasmic region by substituting alanine for the tyrosine at amino acid positions 204 and 211 (US8-Y204/211A; Fig. 5a). Although cells expressing US8-Y204/211A displayed discernible punctate structures, the mutant protein failed to localize to the lysosomes, as evidenced by its colocalization not with LAMP1 (Supplementary Fig. 7a). To examine the effects of US8ΔCT and US8-Y204/211A on antiviral responses induced by TLR3 and TLR4, we analyzed *cxcl10* and *ifnb* mRNA expression levels in cells expressing empty vector, wild-type US8, US8ΔCT, or US8-Y204/211A after stimulation with dsRNA or LPS. Both mutant proteins significantly restored the mRNA expression levels of *cxcl10* and *ifnb* as well as the *ifnb* promoter activity (Fig. 5f and Supplementary Fig. 7b). In contrast to wild-type US8, the US8ΔCT and US8-Y204/211A mutants were incapable of inducing TLR3/TLR4 or UNC93B1 degradation (Fig. 5g and Supplementary Fig. 7c). In addition, the cell-surface expression of TLR4 was restored in cells expressing US8ΔCT or US8-Y204/211A to the extent of being essentially indistinguishable from that in empty-vector-expressing cells (Fig. 5h). Taken together, these findings suggest that the cytoplasmic region of both US7 and US8 is involved in the ability of the viral proteins to destabilize TLR3 and TLR4, and particularly, that the tyrosine motifs within the US8 cytoplasmic region plays a role of transporting US8 and its targets to the lysosome, leading to lysosomal degradation of TLR3 and TLR4 and subsequent suppression of antiviral responses.

**US7 and US8 subvert the innate antiviral response in vivo**. To directly demonstrate the physiological relevance of US7 and US8 in suppressing TLR-mediated antiviral responses, we investigated the functions of US7 and US8 during infection by wild-type HCMV strain AD169. To do that, we used an HCMV deletion mutant lacking the US7-US16 region (HCMVΔUS7-16), which is nonessential for viral replication, and recombinant versions of HCMVΔUS7-16 with reinserted US7 or US8 (HCMVΔUS7-16-Rev.US7 and HCMVΔUS7-16-Rev.US8; Fig. 6a). We initially examined the time courses of US7 and US8 expression in wild-type HCMV-infected HFF cells. The mRNA expression levels of both proteins increased steadily at the early time points and were sustained at later time points during the infection (Fig. 6b, left

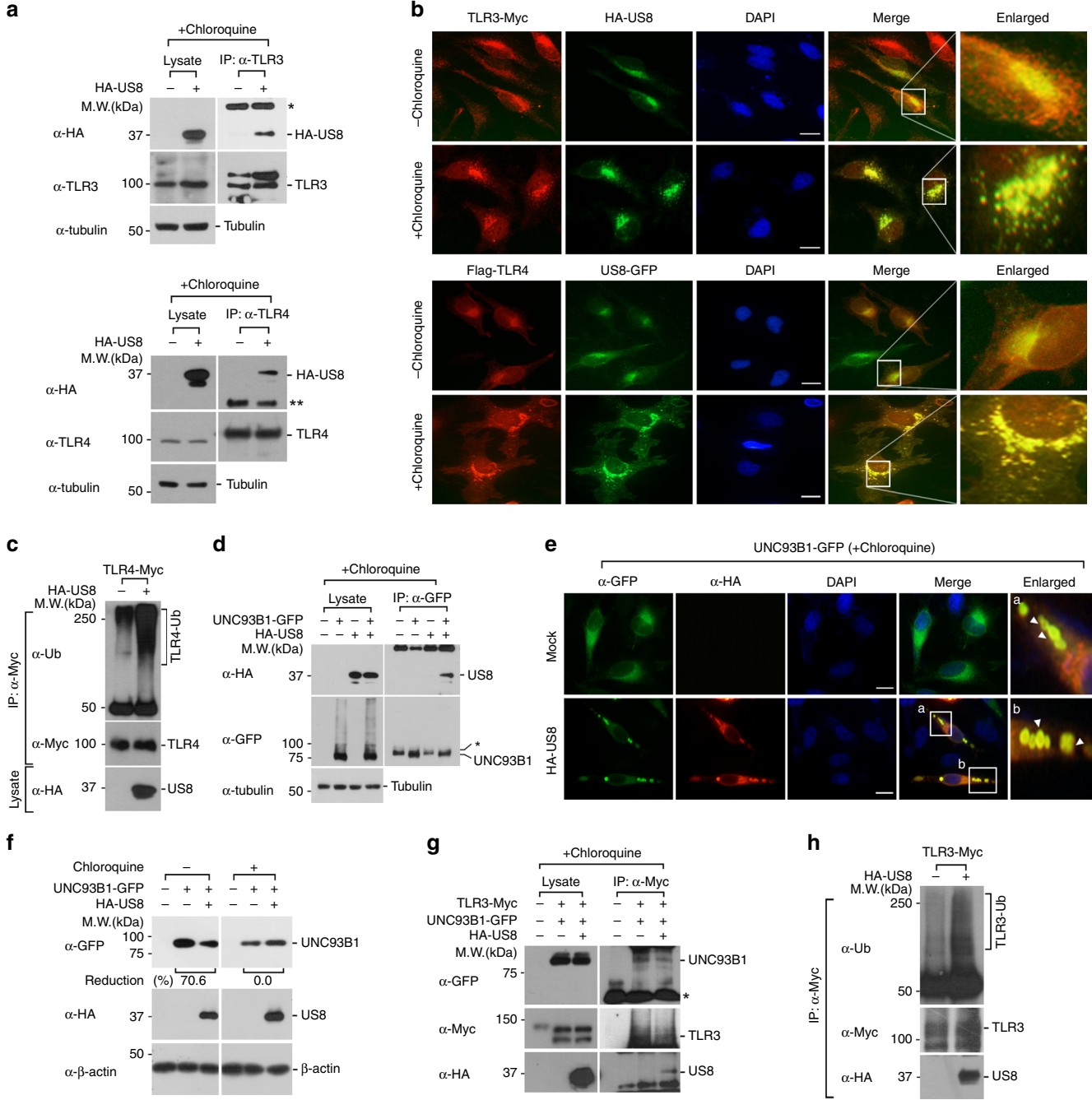

**Fig. 4** US8 promotes TLR3 and TLR4 destabilization by disrupting TLR3-UNC93B1 interaction and targeting TLR4 to lysosomes. **a** US8 interacts with TLR3 and TLR4. HeLa cells were transfected with empty vector or HA-US8 and then treated with 100 μM chloroquine for 4 h. Lysates were immunoprecipitated with anti-TLR3 or anti-TLR4 antibody and immunoblotted with anti-TLR3, anti-TLR4, anti-HA, or anti-Tubulin antibody. *, Ig heavy chains; **, Ig light chains. **b** US8 co-localizes with TLR3 or TLR4. HeLa cells expressing TLR3-Myc or Flag-TLR4/MD-2-Myc were transfected with empty vector, HA-US8 or US8-GFP treated with or without chloroquine, and then stained with anti-Myc, anti-Flag or anti-HA antibody. DAPI was used as a nuclear counterstain. Scale bars, 10 μm. **c** US8 promotes TLR4 ubiquitination. Cells expressing TLR4-Myc with or without HA-US8 were treated with 20 μM MG132 for 4 h and lysed. Cell extracts were immunoprecipitated with anti-Myc antibodies and then analyzed by immunoblotting with indicated antibodies. **d**, **e** US8 associates with UNC93B1. HeLa cells expressing UNC93B1-GFP were transfected with empty vector or HA-US8 and immunoblotted **d** or stained **e** with indicated antibodies. Scale bars, 10 μm. *, non-specific bands. **f** US8 reduces UNC93B1 stability in a lysosome-dependent manner. HeLa cells expressing UNC93B1-GFP were transfected with empty vector or HA-US8 and treated with 100 μM chloroquine for 4 h. Lysates were immunoblotted with the indicated antibodies. **g** US8 disrupts the TLR3-UNC93B1 interaction. TLR3-Myc-expressing HeLa cells were transfected with UNC93B1-GFP and empty vector or HA-US8 and then treated with 100 μM chloroquine. Lysates were immunoprecipitated with anti-Myc and immunoblotted with anti-GFP and anti-Myc or anti-HA antibody. *, Ig heavy chains. **h** US8 promotes TLR3 ubiquitination. Cells expressing TLR3-Myc with or without HA-US8 were treated with 20 μM MG132 for 4 h and lysed. Cell extracts were immunoprecipitated with anti-Myc antibodies and then analyzed by immunoblotting with indicated antibodies. Data are representative of at least three independent experiments. Source data are provided as a Source Data file

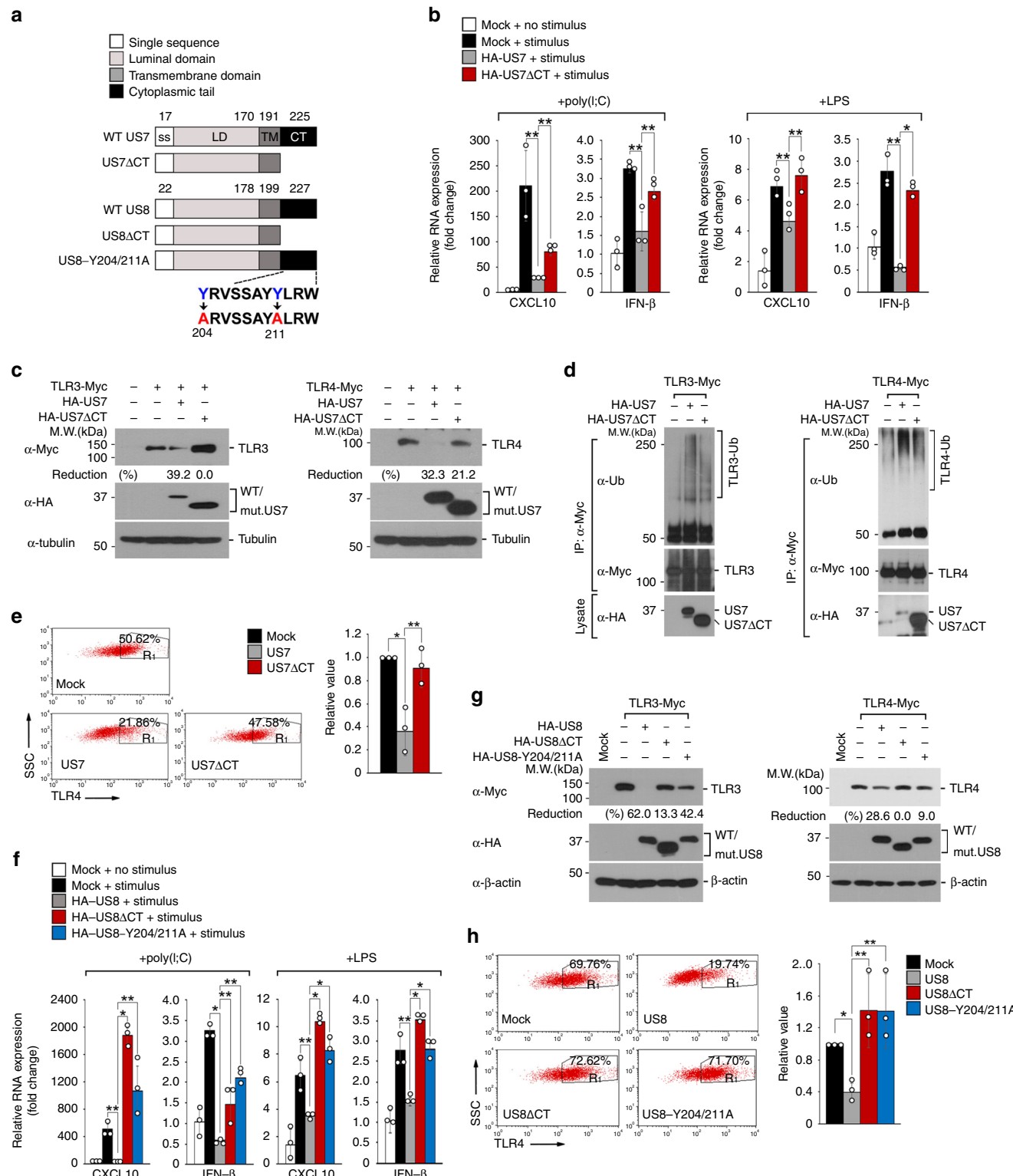

panels). As expected, although HCMVΔUS7-16 successfully infected the host cells, as evidenced by high expression levels of UL44 throughout the course of infection, neither US7 nor US8 was detected during infection by the mutant virus (Fig. 6b, right panels). We further investigated whether US7 or US8 affects TLR3-mediated or TLR4-mediated innate antiviral immune responses in vivo. During wild-type HCMV infection, *ifnb* expression increased at early time points but was gradually

decreased over 48 h, reaching a level that was about 10% of that in cells infected by HCMVΔUS7-16 (Fig. 6c). Whereas the protein levels of TLR3 and TLR4 declined at 12 h and 24 h, respectively, after infection by wild-type HCMV, they were barely affected by infection with HCMVΔUS7-16 (Fig. 6d).

To specify the effects of US7 and US8 on the suppression of antiviral responses mediated by TLR3 and TLR4, we transfected the US7-US16 genes into HCMVΔUS7-16-infected HFF cells and

**Fig. 5** The C-terminal domain of US7 and US8 is required for the function of those proteins. **a** Schematic representation of US7 and US8 and their C-terminal deletion mutants (US7ΔCT and US8ΔCT) and the US8 point mutant (US8-Y204/211A). **b** The effect of US7 C-terminal domain in the blockade of TLR3- or TLR4-mediated *cxcl10* or *ifnb* expression. HeLa cells were transfected with empty vector, wild-type US7, or US7ΔCT and then treated with 10 μg ml$^{-1}$ poly(I:C) or 5 μg ml$^{-1}$ LPS for 12 h. The mRNA expression levels of *cxcl10* and *ifnb* genes were measured by qPCR analysis. *$P < 0.001$, **$P < 0.05$ (Student's *t*-test). **c** US7ΔCT restores TLR3 and TLR4 levels. TLR3-Myc- or TLR4-Myc-expressing HeLa cells were transfected with wild-type US7 or US7ΔCT. Lysates were immunoblotted with anti-Myc antibody. Tubulin was used as a loading control. **d** The involvement of US7 C-terminal domain in the ubiquitination of TLR3 or TLR4. TLR3-Myc- or TLR4-Myc-expressing HEK 293 T cells were transfected with wild-type US7 or US7ΔCT and then treated with 20 μM MG132 for 4 h. Lysates were immunoprecipitated with anti-Myc and immunoblotted with indicated antibodies. **e** US7 CT domain is important for downregulating the cell surface expression of TLR4. Endogenous TLR4 surface expression of US7- or US7ΔCT-expressing U937 was assessed by FACS analysis. Right graph, validation of TLR4 cell surface expression levels in US7- or US7ΔCT-expressing cells by FACS. *$P < 0.001$, **$P < 0.05$ (Student's *t*-test). **f** Both US8ΔCT and US8-Y204/211A are not able to inhibit TLR3- or TLR4-mediated *cxcl10* or *ifnb* expression. HeLa cells were transfected with empty vector, wild-type US8, US8ΔCT, or US8-Y204/211A and then treated with 10 μg ml$^{-1}$ poly(I:C) or 5 μg ml$^{-1}$ LPS for 12 h. *cxcl10* and *ifnb* mRNA levels were measured by qPCR analysis. *$P < 0.001$, **$P < 0.05$ (Student's *t*-test). **g** US8-Y204/211A mutant restores TLR3 and TLR4 levels. TLR3-Myc- or TLR4-Myc-expressing HeLa cells were transfected with wild-type US8, US8ΔCT or US8-Y204/211A. Lysates were immunoblotted with indicated antibodies. **h** US8ΔCT or US8-Y204/211A restores TLR4 surface. Endogenous TLR4 surface expression of wild-type-, US8ΔCT-, or US8-Y204/211A-expressing U937 was assessed by FACS analysis. Bottom graph, validation of TLR4 cell surface expression levels in US8-, US8ΔCT-, or US8-Y204/211A-expressing cells by FACS. *$P < 0.001$, **$P < 0.05$ (Student's *t*-test). Data are representative of three independent experiments and are presented as means ± s.d. in **b**, **e**, **f**, and **h**. Source data are provided as a Source Data file

examined *ifnb* mRNA production. We first observed overexpressed levels of HA-US7 and HA-US8 in HFF cells, confirming that these US genes were successfully overexpressed at levels comparable to that observed during wild-type HCMV infection (Supplementary Fig. 8a). Consistently, we observed *ifnb* mRNA expression at 24 h, but not at 96 h, after wild-type HCMV infection or stimulation with poly(I:C) (Supplementary Fig. 8b, lanes 3 and 4). In contrast, *ifnb* mRNA expression was highly increased at 96 h after infection with HCMVΔUS7-16 (Supplementary Fig. 8b, lanes 4 and 5). The introduction of both US7 and US8 reduced the *ifnb* mRNA levels in HCMVΔUS7-16-infected HFF cells (Supplementary Fig. 8b, lanes 6 and 7). We also observed a considerable reduction of *ifnb* mRNA levels in cells transfected with US10, but not in those transfected with US11-US16 (Supplementary Fig. 8b, lanes 9-15). Similar to the patterns in cells infected with wild-type HCMV, the stability of TLR3-Myc and TLR4-Myc was decreased when US7 or US8 was ectopically expressed in HCMVΔUS7-16-infected HFF cells (Supplementary Fig. 8c).

To exclude the possibility that the transfection system with each US gene had any side effects, we examined endogenous TLR3 and TLR4 protein levels in HFF cells infected with HCMVΔUS7-16-Rev.US7 or HCMVΔUS7-16-Rev.US8. Immunoblot analysis of infected-cell lysates showed that the overall amounts of endogenous TLR3 and TLR4 proteins, but not another membrane protein, Calnexin, decreased considerably after infection with wild-type HCMV, HCMVΔUS7-16-Rev.US7, or HCMVΔUS7-16-Rev.US8, whereas they remained unchanged in cells infected with HCMVΔUS7-16 (Fig. 6e). Consistent with the in vitro results, the mRNA expression levels of *cxcl10*, *cxcl11*, *tnfsf10*, and *ifnb* were severely reduced in HFF cells infected with wild-type HCMV, HCMVΔUS7-16-Rev.US7, or HCMVΔUS7-16-Rev.US8, whereas they were induced normally in cells infected with HCMVΔUS7-16 (Fig. 6f). Overall, our results provide direct evidence for an essential in vivo role of US7 and US8 in subverting the antiviral innate response by targeting TLR3 and TLR4.

## Discussion

Many viruses have been reported to target TLR signaling pathways that activate both innate and adaptive immunity. However, how HCMV blocks TLR-mediated antiviral responses has yet remained unknown. In this study, we report that the HCMV-encoded glycoproteins US7 and US8 target the TLR3 and TLR4 signaling pathways by promoting the degradation of those TLRs, which results in the overall downregulation of the TLR-mediated antiviral immune response. We found that US7 binds to Derlin-1 and Sec61β together with TLR3 or TLR4, leading to the subsequent degradation of those TLRs in a proteasome-dependent manner. Furthermore, US8 not only disrupts the binding affinity of TLR3 for UNC93B1 by a competitive interaction with both TLR3 and UNC93B1, but it also causes TLR4 to be aberrantly transported to the lysosome, eventually leading to the destabilization of both TLRs and the subsequent suppression of antiviral innate immunity (Fig. 7). The expression of TLR3 and TLR4 was significantly downregulated during wild-type HCMV infection, but not during infection by mutant HCMV with a deletion of the US7-US16 region. The loss of TLR3 and TLR4 downregulation by the mutant HCMV was reversed by the introduction of US7 or US8, which provided evidence of the physiological relevance of those viral proteins. Overall, our results identify a previously unknown strategy for evading the innate immune response in which HCMV-encoded glycoproteins US7 and US8 both negatively regulate the antiviral response by targeting TLR3 and TLR4.

Particularly, US7 uses Sec61 and Derlin-1 to promote ubiquitination of TLR3 and TLR4, leading to ubiquitin-dependent degradation in the proteasome, which is similar to US2- or US11-mediated destruction of MHC class I molecules. Several studies propose that US2 appropriates the TRC8 (translocation in renal carcinoma, chromosome 8 gene) E3 ubiquitin ligase to degrade MHC class I molecules, whereas degradation induced by US11 is dependent on the TMEM129 E3 ligase[48,49]. We thus speculate that US7 may possibly interact with TRC8 or TMEM129 E3 ligase, thereby facilitating ubiquitination and subsequent degradation of TLR3 or TLR4.

The functional domain study of US7 and US8 revealed that the C-terminal region of both proteins is essential for the proteins ability to destabilize TLR3 and TLR4. In particular, US8ΔCT completely failed to localize to lysosomes, suggesting that other residues in the C-terminal domain of US8 may be involved in accurately targeting US8 to the lysosome. Due to the defective lysosomal translocation, the ability of the US8-Y204/211A mutant to inhibit TLR3 and TLR4 destabilization was significantly impaired. Although each glycoprotein encoded by the US2-US11 region localizes mostly at the ER, each has a different function in evading the host immune response. That may be due to the diversity of the C-terminal region among the US proteins, which determines the target and

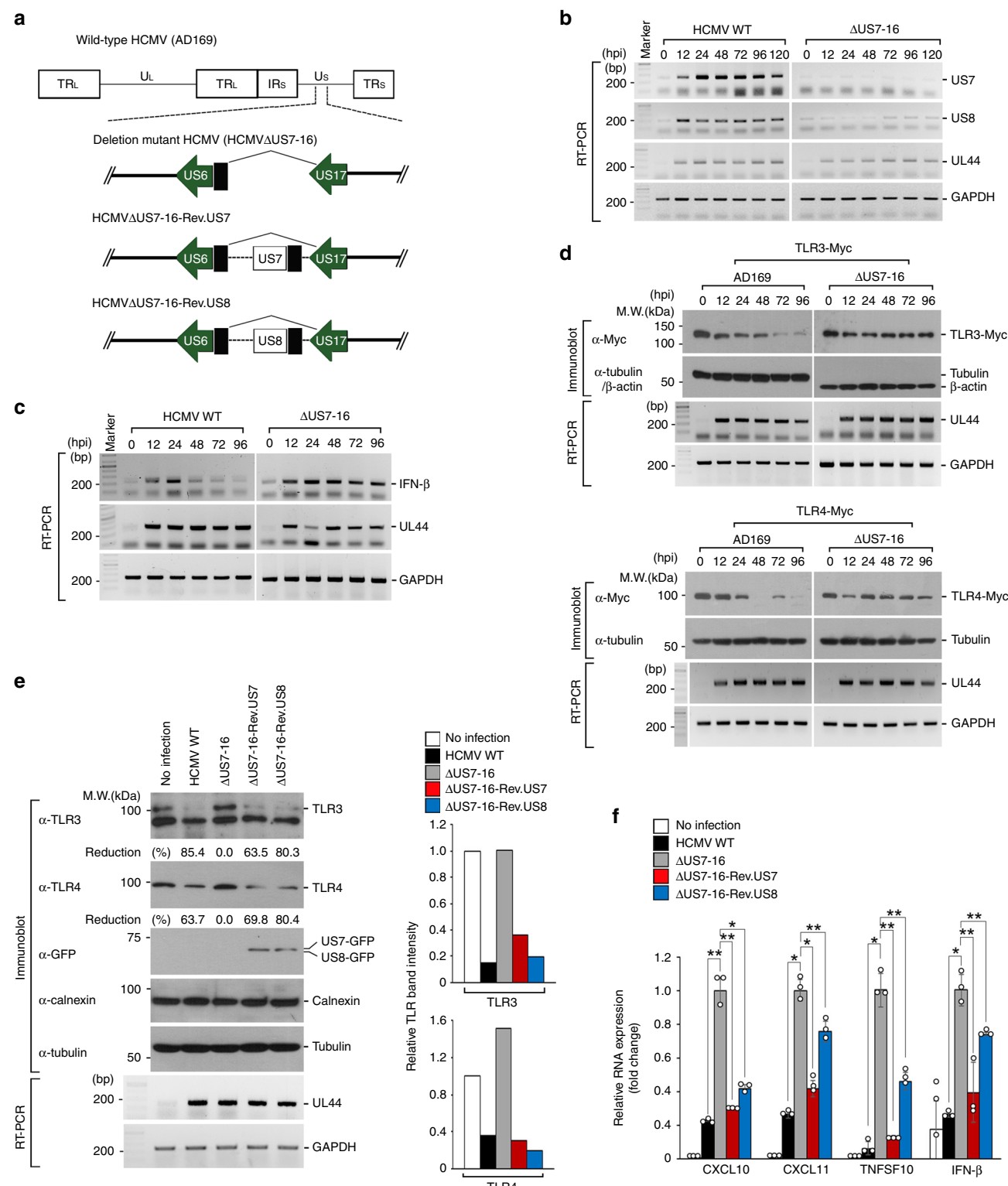

functionality of the proteins in immune evasion. In addition, whether other domains of US7 or US8 may be involved in the antagonistic capabilities of those proteins needs to be better understood.

Interestingly, HCMV miRNAs selectively block host immune responses to viral infection[12,37,50,51]. It is also proposed that HCMV regulates the expression of one of its own genes, US7, by two different miRNAs, miR-US5-1 and miR-US5-2, thus down-regulating US7 mRNA expression late in the infection[52]. The

mRNA expression level of US7 was slightly decreased after 96 h of HCMV infection (Fig. 6b), which may help to prevent apoptosis caused by excessive viral burden, thus facilitating a persistent infection. Indeed, HCMV is characterized by a protracted replication cycle and the blocking of cellular apoptosis or necrosis, which keep the host cell alive[53].

Several lines of evidence suggest that HCMV enhances intestinal bacterial infections, such as Salmonella typhimurium infections[54,55]. In addition, the prevalence of HCMV infection in

**Fig. 6** US7 and US8 suppress innate antiviral response in vivo. **a** Schematic representation of the HCMV strains used. HCMV WT, wild-type HCMV AD169; HCMVΔUS7-16, HCMV deletion mutant lacking the US7-US16 region; HCMVΔUS7-16-Rev.US7 or -Rev.US8 HCMV mutant lacking all genes in the US7-US16 region restored with US7 or US8 expression. **b, c** HCMV, but not HCMVΔUS7-16, blocks antiviral response through destabilizing TLR3 and TLR4 proteins. HFF cells were infected with HCMV WT or HCMVΔUS7-16 at an MOI of 2 for the indicated times. The presence of HCMV-derived US7 or US8 and *ifnb* mRNA expressions were analyzed by RT-PCR. UL44 and GAPDH were used as HCMV infection and loading controls, respectively. **d** HCMV, but not HCMVΔUS7-16, degrades TLR3 and TLR4 proteins at late times. TLR3-Myc- or TLR4-Myc-expressing HFF cells were infected with HCMV WT or HCMVΔUS7-16 and the lysates were immunoblotted with indicated antibodies. RT-PCR analysis of UL44 and GAPDH was used as HCMV infection and loading controls, respectively. **e** US7 and US8 decreases TLR3 or TLR4 expression levels in vivo. HFF cells were infected with HCMV WT, HCMVΔUS7-16, HCMVΔUS7-16-Rev.US7, or HCMVΔUS7-16-Rev.US8. Lysates were immunoblotted with anti-TLR3, anti-TLR4, anti-GFP, anti-Calnexin, or anti-Tubulin antibody. RT-PCR analysis of UL44 and GAPDH was used as HCMV infection and loading controls, respectively. The intensity of TLRs bands was quantified as comparing the relative abundance of TLR3 or TLR4 to Tubulin (right graphs). **f** Both US7 and US8 block innate antiviral response in vivo. HFF cells were infected with HCMV WT, HCMVΔUS7-16, HCMVΔUS7-16-Rev.US7, or HCMVΔUS7-16-Rev.US8. The indicated gene expression was measured by qPCR. $*P < 0.001$, $**P < 0.05$ (Student's $t$-test). Data are representative of three independent experiments and are presented as means ± s.d. in **f**. Source data are provided as a Source Data file

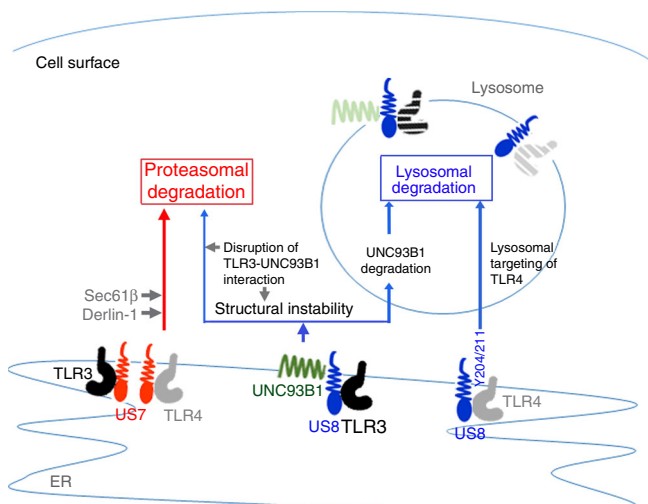

**Fig. 7** Proposed model of the novel function of HCMV US7 and US8 in the inhibition of TLR3- or TLR4-mediated innate antiviral response. Schematic model of US7- or US8-mediated negative regulation of antiviral response by targeting TLR3 and TLR4

the colon is significantly higher in patients with inflammatory bowel disease (IBD) relative to that in controls, supporting the hypothesis that IBD severity is positively linked to HCMV infection[56,57]. Although the mechanisms of synergistic pathogenesis in viral-bacterial coinfections have remained elusive, on the basis of published studies and our findings here, it is plausible that HCMV infection may directly contribute to increased susceptibility to intestinal invasive bacteria by selectively targeting TLR3 and TLR4 and, furthermore, may be a considerable risk factor for developing the severity of secondary bacterial infections or IBD.

For HCMV to establish a latent infection, it has adopted many strategies to block IFN and cytokine production throughout the course of infection. Previous studies suggest that HCMV encodes several different genes, such as *ul44*, *ul82*, *ul83*, *ul122*, and *ul123*, which are all essential for inhibiting IFN production[17,58–62]. Additionally, because TLRs play a pivotal role in the activation of both innate and adaptive immunity during viral infection, our findings support the hypothesis that HCMV has also evolved strategies to block IFN production by targeting TLR proteins. That suggests the possibility of redundancy among the viral genes that obscures important functions of the individual antagonists in the suppression of IFN responses under different conditions. That would explain why the deletion of the US7-US16 region in HCMV did not result in attenuated replication of the virus. In

other words, the HCMVΔUS7-16 mutant virus might use other viral proteins; such as UL44, UL82, UL83, UL122, or UL123; to rescue the viral blockade of the IFN response in place of US7 and US8, thus maintaining viral replication in the host. Given that the HCMV-encoded glycoproteins US7, US8, and US9 can inhibit the innate antiviral response, and also that US10 seems to attenuate IFN-β production, the US7-US10 region of the HCMV genome might be a reservoir of viral genes that function together and simultaneously to subvert host innate antiviral responses, particularly at later time points of infection. Our findings provide the first molecular mechanism by which an HCMV-encoded glycoprotein blocks a TLR-mediated signaling pathway, which will improve the understanding of HCMV pathogenesis and support future development of potential therapeutics for HCMV-associated diseases.

## Methods

**Cell lines**. Human macrophages cell line U937 (CRL-1593.2, ATCC, Manassas, VA) and human monocyte cell line THP-1 (TIB-202, ATCC) were cultured in RPMI1640 supplemented with 10% heat inactivated fetal bovine serum (FBS) (HyClone, Logan, UT) and penicillin/streptomycin (Hyclone). Human embryonic kidney (HEK) 293 T (CRL-3216, ATCC), HeLa (CCL-2, ATCC), and human foreskin fibroblasts (HFFs) (SCRC-1041, ATCC) cells were cultured in DMEM supplemented with 10% heat inactivated FBS and penicillin/streptomycin. Cells were grown at 37 °C in humidified air with 5% $CO_2$.

**DNA constructs**. HCMV US7 and US8 constructs were fused at the N-terminus to a hemagglutinin (HA) tag containing N-terminal H2-K$^b$ signal sequences. We then generated the C-terminal deletion constructs of HA-US7 (HA-US7ΔCT) and HA-US8 (HA-US8ΔCT) by PCR and the point mutant HA-US8 (HA-US8 Mut; Y204A/Y211A) by site-directed mutagenesis. All of the N-terminal HA-tagged US7 and US8 or C-terminal HA-US12-US16 genes into pcDNA3.1 or pMSCV, respectively (Invitrogen, San Diego, CA). In addition, HA-US7 and HA-US8 were fused at the C-terminus to green fluorescence protein (GFP) by subcloning them into a pEGFP N3 vector (Clontech, Palo Alto, CA). We also sub-cloned all of the above constructs, including TLR4-Myc, TLR3-Myc, MD2-Myc, Flag-TLR4, UNC93B1-GFP, into a retroviral vector pMSCV (Clontech) or a pLHCX vector (Clontech). We verified all constructs by sequencing. All primer sequences are listed in Supplementary Table 1.

**Antibodies and reagents**. The following antibodies were used: HA (G036, ABM Inc., Richmond, Canada, 1:1000), Myc (2276, Cell Signaling Technology, Danvers, MA, USA, 1:1000), Flag (M185-3L, MBL International, Woburn, MA, USA, 1:1000), GFP (ab290, Abcam, Cambridge, UK, 1:1000), Tubulin (G094, ABM Inc., 1:2000), human TLR3 (sc-32232, Santa Cruz Biotechnology, Santa Cruz, CA, USA; ab62566, Abcam, Cambridge, UK, 1:1000), human TLR4 (sc-293072, Santa Cruz Biotechnology; 312802, BioLegend, San Diego, CA, USA, 1:1000), LAMP1 (ab24170, Abcam, 1:1000), PDI (#ADI-SPA-891, Enzo Life Sciences, Farmingdale, NY, USA, 1:100), GM130 (610822, BD Biosciences, Mountain View, CA, USA, 1:100), Ubiquitin (sc-8017, Santa Cruz Biotechnology, 1:1000), Alexa Fluor 488 anti-mouse (A-11029, Life Technologies, Carlsbad, CA, USA, 1:100), and Alex Fluor 568 anti-rabbit (A-11036, Life Technologies, 1:100). Antibodies to Sec61β/Derlin-1 and Calnexin/MHC class I molecules were kindly provided by Dr. Hidde L. Ploegh (Boston Children's hospital, Harvard Medical School, MA, USA, 1:1000) and Dr. Kwangseog Ahn (Seoul National University, Seoul, South Korea, 1:1000),

respectively. The following reagents were used: LPS (L3755, Sigma-Aldrich, St Louis, MO, USA), poly(I:C) (tlrl-pic, InvivoGen, San Diego, CA, USA), Pam3CSK4/Imiquimod (InvivoGen, San Diego, CA, USA), CpG-DNA (TIB Mol-biol, Berlin, Germany), Endo H (P0703, New England Biolabs, Ipswich, MA, USA), MG132 (M-1157, AG Scientific, San Diego, CA, USA), Chloroquine (C6628, Sigma-Aldrich), DAPI (4′,6-diamidino-2-phenylindole, D9542, Sigma-Aldrich).

**Viruses**. HCMV wild-type AD169 and mutant AD169ΔUS7-16, in which US7-16 region is deleted and replaced with 48 bp FRT site. Recombinant HCMVΔUS7-16-Rev.US7/US8 used in the study was constructed utilizing a recombination strategy and the materials were provided by Dr. Jun-Young Seo. Briefly, the Tet-on inducible US7- or US8-GFP fusion plasmid was inserted into mutant HCMV AD169ΔUS7-16 genome maintained in a bacterial artificial chromosome (BAC) using a recombination strategy. *Escherichia coli* (*E. coli*) SW105 strain, carrying BAC of mutant HCMVAD169ΔUS7-16 and *ara*-inducible *Flpe* recombinase gene, was transformed with pO6-SVT-entry plasmid containing US7- or US8-GFP. The *ara*-induced FLP recombinase induces site-specific recombination between the FRT sequences in the BAC and the insertion plasmid. To reconstitute the recombinant BAC containing US7- or US8-GFP into viruses, HFF cells were electroporated and were incubated until the recombinant HCMVΔUS7-16-Rev.US7/US8 viruses were generated. Virus stocks were prepared by infecting HFF cells with (MOI = 0.01) and incubating until 100% of cells showed cytopathic effects. Then cells were scraped and HCMV particle containing cell pellet and the supernatant were collected. The viral stocks were distributed in small aliquots, and stored at −80 °C. Virus stock aliquots were freshly thawed each time and not reused to avoid defective viral particles.

**HCMV infection**. HFF cells were plated in six-well plates and cultured in DMEM until cells were 70-80% confluent. HCMV strains were infected at a multiplicity of infection (MOI) as indicated. After incubating the infected cells for the indicated number of dpi, cells were washed and re-fed with fresh DMEM. Virus-infected cells were incubated for indicated time periods as indicated. To determine HCMV infectivity in all infection experiments, HCMV-infected HFF cells were stained with an anti-IE1 antibody and quantified by measuring viral infectivity.

**Transfection and retroviral transduction**. Cells were transfected using Oimcsfect (Omicsbio, Taipei City, Taiwan) in serum-free and antibiotic-free DMEM for 20-36 h. For the maximal transfection efficiency of HFF cells, calculate the number of HFF cells plated to obtain 70-80% confluence and incubate the cells at 37 °C in CO2 incubator for 24 h before transfection. All cell lines were tested for mycoplasma and were confirmed free of contamination. For preparation of virus particles, HEK 293 T cells were transfected with plasmids encoding VSV-G and Gag-Pol, together with constructs cloned into retroviral vector containing target gene. Virus-containing supernatants were obtained at 48 h post-transfection and filtered through a 0.45-µm filter. Cells were transduced with virus by centrifugation at 2,200 rpm for 45 min, and then incubated for 4 h. Transduced cells were incubated with fresh media for 24 h and then selected with puromycin.

**Enzyme-linked immunosorbent assay**. THP-1 cells were stimulated with 5 µg ml$^{-1}$ poly(I:C) or 10 µg ml$^{-1}$ LPS for 24 h. Cell culture supernatants were collected, and human IFN-β or IL-6 levels were analyzed by ELISA assay according to the manufacturer's recommendations (IFN-β: #41410-1, PBL Assay Science, NJ, USA, IL-6: BD Biosciences, San Jose, CA).

**Flow cytometry**. The expression levels of TLR4 on the cell surface were determined by flow cytometry (FACScalibur, BD Biosciences) after indirect immuno-fluorescence using anti-TLR4 antibody and fluorescein isothiocyanate (FITC)-conjugated secondary antibodies.

**Pulse-chase experiment and immunoprecipitation**. Cells were starved in methionine/cysteine free DMEM for 1 h prior to pulse-labeling for 1 h using [$^{35}$S] methionine/cysteine at 0.1 mCi ml$^{-1}$ (EasyTag Express $^{35}$S Protein labeling mix, Perkin Elmer, Waltham, MA, USA) for 1 h at 37 °C. The labeling cells were chased for 4 h with DMEM media containing 10% FBS. After once wash with phosphate-buffered saline (PBS), cells were lysed using 1% Nonidet P-40 (NP-40) in PBS supplemented with protease inhibitors for 30 min. Lysates were incubated with primary antibodies and followed by protein G-Sepharose beads at 4 °C. The beads were then washed three times with 0.1% NP-40/PBS and proteins were eluted by boiling the samples in denaturing buffer. Eluted proteins were treated with Endo H (New England Biolabs) and separated by sodium dodecyl sulfate (SDS)-poly-acrylamide gel electrophoresis (PAGE). The gels were dried, exposed to BAS film, and analyzed by Phosphor Imaging System BAS-2500 (Fuji Film Company, Tokyo, Japan).

**Co-IP and immunoblot analysis**. Cells were lysed with 1% NP-40 (for TLR3/4-US7/8 or UNC93B-US8 interactions) or 1% digitonin (for Sec61β/Derlin-1-US interactions) with protease inhibitors. The lysates were then incubated with primary antibodies followed by protein G-Sepharose beads at 4 °C. The beads were

then washed three times with 0.1% NP-40 or 0.1% digitonin. The proteins were eluted by boiling the samples, or heating the samples without boiling (for the UNC93B1-US8 interaction), in denaturing buffer. Protein samples were separated by SDS-PAGE and transferred to polyvinyl difluoride (PVDF) membrane (Milli-pore, Bedford, MA, USA). The membranes were blocked with 5% skim milk in PBS containing 0.1% Tween 20 (PBS-T) for 10 min and incubated with the appropriate antibodies at 4 °C overnight. The membranes were washed three times with PBS-T and incubated with horseradish peroxidase (HRP)-conjugated secondary anti-bodies for 1 h. Bands were visualized using an enhanced chemiluminescence (ECL) detection reagent (Advansta, Menlo Park, CA, USA). All the original blots images are provided as the Source Data file.

**Ubiquitination assay**. For ubiquitination assay, cells transiently expressing indi-cated plasmids were incubated for 20–24 h. Cells were harvested and lysed in RIPA buffer (50 mM Tris-HCl (pH 8.0), 150 mM NaCl, 1% NP-40, 0.1% SDS, and 1 mM EDTA) containing protease inhibitor cocktail and 10 µM deubiquitinase inhibitor N-ethylmaleimide (NEM, Sigma). The cell lysate was immunoprecipi-tated with anti-Myc antibody overnight at 4 °C and then protein G-Sepharose beads were added to the samples for 1-1.5 h at 4 °C. The beads were washed three times with RIPA buffer and proteins were eluted by boiling in 1 × SDS loading buffer. Analysis of ubiquitination was performed by immunoblotting using anti-Ub antibody.

**RT-PCR and qPCR**. Total cellular RNA was prepared using an RNA prep kit (GeneAll, Seoul, South Korea). RNA (0.5 µg ml$^{-1}$) was reverse transcribed with oligo(dT) primers at 42 °C for 1 h using Moloney Murine Leukemia Virus (M-MLV) reverse transcriptase (Enzynomics, Daejeon, South Korea). PCR products were visualized on ethidium bromide-stained gels. The qPCR reactions were per-formed on QuantStudio 3 Real-Time PCR system (Applied Biosystems, Foster City, CA, USA) using SYBR Green (Enzynomics). Glyceraldehyde 3-phosphate dehy-drogenase (GAPDH) was used for normalization. All primer sequences are listed in Supplementary Table 1.

**Immunofluorescence assay**. For immunofluorescent staining, cells were fixed in 3.7% formaldehyde and permeabilized with 0.1% Triton X-100. After blocking with 2% bovine serum albumin (BSA) in PBS (PBA) for 30 min, the samples were incubated with the appropriate primary antibody in 2% PBA for 1 h at room temperature. Bound antibody was visualized with an Alexa Fluor 488- or Alexa Fluor 568-conjugated secondary antibody (Life Technologies) by fluorescence microscope. DAPI was used as a nuclear counterstain. The fluorescence intensity (FI) of the IFA images was quantified using the Zen software (Carl Zeiss) (http://zeiss.com). At least four randomly chosen fields for a total of at least 30 cells were analyzed. The FI is given in arbitrary units as an average value per cell in the selected representative fields.

**Luciferase assay**. Cells in 12 well plate were transfected with NF-κB or IFN-β firefly luciferase, Renilla luciferase, along with empty vector, US7 or US8. The transfected cells were then stimulated with poly(I:C) or LPS and lysed with lysis buffer. The luciferase activity was determined using the Dual-Luciferase Reporter Assay System (Promega, Madison, WI, USA). Firefly luciferase activity was nor-malized to Renilla luciferase activity.

**Microarray analysis**. Total RNA was isolated using an RNeasy Mini kit (QIAGEN, Hilden, Germany) and the integrity of the RNA was evaluated using an ND-1000 Spectrophotometer (NanoDrop, Wilmington, DE, USA) and Agilent 2100 Bioa-nalyzer (Agilent Technologies, Palo Alto, CA, USA). We executed the Affymetrix whole transcript expression array process according to the manufacturer's protocol (GeneChip Whole Transcript PLUS reagent Kit). cDNA was synthesized using the GeneChip WT (Whole Transcript) Amplification kit according to the manu-facturer's instructions. We then fragmented the sense cDNA and biotin-labeled it with terminal deoxynucleotidyl transferase using the GeneChip WT Terminal Labeling kit. Approximately 5.5 µg of labeled DNA target was hybridized to the Affymetrix GeneChip Human Gene ST 2.0 ST Array at 45 °C for 16 h. We then washed the hybridized arrays, stained them using a GeneChip Fluidics Station 450, and scanned them on a GCS3000 Scanner (Affymetrix). Signal values were com-puted using the Affymetrix® GeneChip™ Command Console software. For analysis, raw data were extracted automatically using the Affymetrix data extraction pro-tocol and the software provided by Affymetrix GeneChip® Command Console® software (AGCC). After importing CEL files, we summarized all of the data and normalized it using the robust multi-average method implemented in the Affy-metrix GeneChip® Console™ software (EC). Statistical significance of the expression data was determined using the fold change. To identify differentially expressed genes, we performed hierarchical cluster analysis using complete linkage and Euclidean distance as a measure of similarity. All data analysis and visualization of differentially expressed genes was conducted using R 3.1.2 (www.r-project.org).

**Statistical analysis**. All experiments were repeated at least three times with consistent results. Data are presented as mean ± s.d. (as noted in figure legends).

Statistical differences between two means were evaluated with the two-tailed unpaired Student's *t*-test. Differences with *P* values below 0.05 were considered significant. Presented data were normally distributed and the variances were similar between the groups being statistically compared. No statistical method was used to predetermine sample sizes. Sample size was based on previous experience with experimental variability. No samples were excluded from the analysis. The experiments were not randomized. The investigators were not blinded to allocation during experiments.

**Reporting summary**. Further information on research design is available in the Nature Research Reporting Summary linked to this article.

## Data availability

The raw data of microarray is available in the Gene Expression Omnibus database under the accession number GSE136835. The raw data underlying Figs. 1d–g, 2a,b, 2d-e, 3a, 3c–e, 4a, 4c–d, 4f–h, 5b–h, 6b–f, as well as Supplementary Figs. 1b, 2a–c, 3a–d, 5a,b, 6b,c, 7b,c, and 8a–c are available in the Source Data file. All other data generated or analyzed during this study are available from the corresponding author on reasonable request.

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

## Acknowledgements

This study was supported by grants from the Basic Science Research Program through the National Research Foundation of Korea (NRF) funded by the Ministry of Science, ICT, and future planning (NRF-2016R1A5A1010764 and NRF-2017R1E1A1A01074135).

S.L. was supported by Basic Science Research Program through the National Research Foundation of Korea (NRF) funded by the Ministry of Education (NRF-2018R1D1A1B07048930) and a research grant from the National Cancer Center of Korea (NCC-1710210). A.P., E.A.R., H.J.C., T.A.L., E.L., and S.K. were supported by the Brain Korea (BK21) PLUS Program.

## Author contributions

A.P., E.A.R., H.J.C., T.A.L., E.L., and S.K. conducted experiments and analyzed data. A.P., S.L., J.Y.S., and B.P. designed experiments and wrote the manuscript.

## Competing interests

The authors declare no competing interests.
