## [Peer Review File · Nature Communications]

Reviewers' Comments:

Reviewer #1:

Remarks to the Author:

The manuscript by Park et al. demonstrates that the US7 and US8 glycoproteins of HCMV inhibit innate immunity by targeting TLR3 and TLR4. The role of these two viral proteins in innate signaling has not been demonstrated previously, and thus this manuscript is of importance and is of general interest. While overall the authors provide a convincing case that US7 and US8 inhibit these TLRs, I do have some concerns about the data quantitation, the number of cell lines used in these studies, and some other key controls. I outline some major and more minor concerns that need to be addressed to render this manuscript suitable for publication in *nature Communications*.

Major concerns:

1. Much of the cell line data relies on overexpression of US7 and US8, but it is not stated how strongly these genes are overexpressed. This should be monitored, and optimally, these genes should be overexpressed at levels comparable to that observed during HCMV infection.
2. Many studies throughout the manuscript need quantitation and further statistical analysis. For example, the authors could quantitate the overlap among GFP fusions and organelle markers in Supplementary Figure 1 (and other figures), the protein levels normalized to appropriate housekeeping genes for the blots in Figure 2A and Supplementary Figure 1C, and so on.
3. It is a little disconcerting that the authors perform various studies in at least five different cell lines (HFF, HELA, HEK293s, THP1s, and U937s). While there are advantages to working in the different systems for different assays, it is not explained why particular studies were performed in particular systems. This should be addressed.
4. Much of the analysis of TLR3 and TLR4 is done with tagged proteins. The authors should repeat some of these studies with the endogenous TLRs. For example, can the authors observe US7 and US8 mediated destruction of the endogenous TLR proteins (Figure 2A) or demonstrate the effect of Sec61b/Derlin1 on endogenous TLR proteins (Figure 3D,E)?

Minor concerns:

1. Lines 107 and 367. I'd rephrase this to avoid ambiguity and state that HCMV diminishes TLR3 and TLR4 protein levels (not expression, which refers to gene expression).
2. Line 124. While it is explained later, I'd add to this sentence that the changes in gene expression were monitored using microarrays.
3. Supplementary Figure 1 seems like a grab-bag of several studies. It might be preferable to split this into a localization supplemental figure and a signaling supplemental figure. At a minimum, this figure title could be altered.
4. Are the mRNA levels for TLRs in Supplemental Fig. 1C from the same study using the overexpressed myc-tagged proteins in Fig 2A? If so, this needs need to be clarified in Supplemental Fig 1C. If not, then why not?
5. Line 177. I'd delete "maturation" since this was not studied.
6. The effects of Sec61b and Derlin1 shRNA seem somewhat mixed. Are the observed differences statistically significant?
7. As the authors note, TLR4 localization will be affected by stimulation with LPS for example. It would be worth re-examining the effect of US8 on TLR4 localization in stimulated cells.
8. Line 266. I'd rephrase "protein turnover rate" since the authors aren't measuring this. Maybe protein accumulation?
9. The control data for US7dCT that demonstrate that the truncated protein is there at comparable levels to the US7 full length protein (Supplemental Figure 2) displays GFP tagged constructs but the

data in Figure 5B and Sup Figure 3C uses HA-tagged constructs. The protein level control should be performed using the same protein and optimally the same sample (the authors do provide this control for the other panels in Fig. 5).

10. The change in TLR4 ubiquitination in Fig. 5D doesn't look very profound.

11. In light of the authors' observation that US8dCT doesn't localize properly, I'd be more cautious in how they interpret the role of the C-terminal domain in the function of that protein.

12. Lines 524-531. Please provide further details about the microscopy. On what system and with what software were the images captured, how were they processed and analyzed. How was it decided/quantitated whether proteins co-localized or not.

Reviewer #2:

Remarks to the Author:

The manuscript entitled "HCMV-encoded US7 and US8 act as antagonists of innate immunity by distinctively targeting TLR signaling pathways" by Park et al. describes a potential function of the unique short genes US7 and US8 to down regulate the TLR3 and TLR4 proteins. In general, the study analyzed TLR3 and TLR4 protein levels in cell systems that over-express wild type and mutant versions of US7 and US8. In addition, viral variants lacking US7 and US8 as well as rescued variants that express US7 and US8 only were analyzed for TLR3 and TLR4 stability and induction of inflammatory factors in virus infected cells. The manuscript describes some novel findings with regards to the function of US7 and US8 to modulate factors important for the innate immune response. However, there are some major issues with the manuscript that include a lack of controls that determine the specificity of US7 and US8 function, inconsistencies among the data with regard to the impact US7 and US8 have on TLR3/4 levels, the experiments are based on over-expressed cell systems, and the mechanism of US7 and US8 function to modulate innate immune regulators during the CMV life cycle is not sufficiently supported by experiments. Some of the specific items are outlined below.

1) Figure 1 nicely demonstrates that the expression of US7 and US8 modulates some of the inflammatory factors upon addition of dsDNA, polyIC, and LPS. However, the rationale to use polyIC and LPS is not clear given that polyIC and LPS are unlikely PAMPS for TLRs during a virus infection. In addition, the authors should include controls with cells that express other US genes that do not impact the inflammatory response.

2) In Figure 2, the down regulation of TLR3 and TLR4 is not very convincing and varies among the different panels. These experiments would greatly benefit from inclusion of controls that include cells that over-express other US genes and analysis of other membrane proteins such as MHC class I or transfected CD4. In addition, the radiolabeled studies have different exposures among the panels. The mature form of TLR4 can be observed in all conditions, despite the conclusion of the text.

3) The experiments in Figure 3 proposing that Derlin1- and Sec61beta are involved in US7 and US8 function to down regulate TLR3/4 is not very convincing. The lack of these genes had minimal impact on the levels of TLRs. Also, the inclusion of analyzing other membrane proteins as controls is lacking.

4) The rationale and results of Figure 4 that examines UNC93B1 as the mechanism of US8 function is over interpreted.

5) As with previous Figures, Figure 5 analyzing the US7 and US8 mutants did not have a major impact on TLR3/4 levels. Also, the data was over interpreted and does not support the manuscript's conclusions.

6) Figure 6 is an important experiment that would provide relevance to the function of US7 and US8. The data support that a CMV infection decreases the levels of over-expressed TLR3 and TLR4. Yet, it would be quite important to examine endogenous levels of TLR3 and TLR4 as well as the protein expression of US7 and US8. In addition, the experiments should include the analysis of a non-specific

cellular protein such as transferrin receptor or exogenously expressed CD4. Another issue was that the levels of TLR3 and TLR4 are inconsistent with the levels of inflammatory markers following a CMV infection.

Reviewer #3:

Remarks to the Author:

Human cytomegalovirus (HCMV) encodes a large number of genes that subvert host immune response. Within a cluster of immune evasion genes in the US region of the HCMV genome are two, US7 and US8, whose function has been unknown. Park and colleagues now report their discovery that these two genes bind to and degrade TLR3 and TLR4. They provide some evidence that US7 acts by promoting degradation of Sec61b and Derlin-1 and that US8 disrupts the TLR3-UNC93B1 interaction and promotes proteasomal and lysosomal targeting and degradation. They show the cytoplasmic tails of each protein are necessary for their phenotypes and importantly, that these proteins function in the context of viral infections.

Some data concerning the details of the mechanisms are not entirely convincing, but overall the studies have been well done, are quite extensive, and support that main, new and interesting conclusions.

Major comments:

1. The authors start this work by demonstrating that US7 and US8 repress the response to transfected dsDNA. If it is true that these genes function is to block TLR3 and TLR4 (sensors of dsRNA and of LPS, respectively) why do they work to block dsDNA?
2. Several figures lack appropriate controls. For example, in Fig 3A, there is no control for nonspecific pull-down of US7. The two control lanes did not have any US7 in the input so of course there will be none in the pull down. What is needed is a control in which HA-US7 is transfected with an myc-tagged irrelevant control protein. This same argument applies to Fig 4A and 4D. Also, it would help to state how much of the lysate was loaded relative to the amount used for the IPs.
3. Line 204, Fig. 3C. Derlin-1 appears to be pulled down by anti-HA even in the cell without either US11 or US7, undermining the interpretation that Derlin-1 associates with US7.
4. Fig 3, D and E. How was US7 introduced into these cells? The impact of knocking down Derlin-1 on TLR4 stabilization is not convincing – 33.9 vs 28.3 is not a reliable difference in immunoblot assays. This result further weakens the argument that US7 acts through Derlin-1.
5. Fig. S5. These results are hard to understand. Transfection of HFF is notoriously inefficient, so only a small fraction of cells is likely to be expressing the US7-10 genes. Therefore, the majority of cells that did not express the genes and were infected with the US7-16 genes would be expected to produce a large amount of interferon. Similarly, panel B results seem implausible. How do the authors account for these considerations?

Minor comments.

1. Line 786. "either 20 uM MG132 for 4 hours" or what?
2. The data in Fig. 4E do not show very convincing evidence for co-localization of UNC93B1 with US8, contrary to the text in line 244-5.

3. Line 266-268. The data in Fig S2A and S2B show quite a marked difference in the amount of the wt vs the deltaCT US7 and US8 under non-reducing conditions. How do the authors interpret this observation?
4. Supplemental fig 3. The legend refers to transfection with empty vector in panel A, but no images are shown.
5. Line 310-311. It is not clear why inactivating the DNA sensor antagonist US9 was necessary.
6. The source or construction of the HCMV recombinants depicted in Fig. 6A should be described in the methods.

Comments from the reviewers that required a response are in bold italics, with each reply appearing in normal font just below the comment. In the replies, new material is emphasized in bold.

Reviewer #1 (Remarks to the Author):

The manuscript by Park et al. demonstrates that the US7 and US8 glycoproteins of HCMV inhibit innate immunity by targeting TLR3 and TLR4. The role of these two viral proteins in innate signaling has not been demonstrated previously, and thus this manuscript is of importance and is of general interest. While overall the authors provide a convincing case that US7 and US8 inhibit these TLRs, I do have some concerns about the data quantitation, the number of cell lines used in these studies, and some other key controls. I outline some major and more minor concerns that need to be addressed to render this manuscript suitable for publication in nature Communications.

Major concerns:

1. Much of the cell line data relies on overexpression of US7 and US8, but it is not stated how strongly these genes are overexpressed. This should be monitored, and optimally, these genes should be overexpressed at levels comparable to that observed during HCMV infection.

We thank the reviewer for pointing out that we need to state how strongly the US genes were overexpressed. We have added more detailed information regarding the overexpression experiment in the Methods section. We have also included new data from immunoblot experiments with anti-GFP or anti-HA antibody showing the overexpression levels of GFP-tagged or HA-tagged US7 and US8 during HCMVΔUS7-16 infection (**new Fig. 6e** and **new Supplementary Fig. 8c**). In addition, we performed an RT-PCR experiment to show HA-tagged US7 and US8 overexpression in HFF cells, confirming that these US genes were successfully overexpressed at levels comparable to that observed during wild-type HCMV infection (**new Supplementary Fig. 8a**).

We have also included data in the revised figures showing the overexpression levels of US7 and US8 in transfected cells determined by RT-PCR, immunoblot, and IFA analysis (**new Figs. 1d, 1f, and 1g**, together with Figs. 2a, 2d, 2e, 3a~3e, 4a~4h, 5c-5d, 5g, Suppl. Figs. 3a, 4~5, 6a-b, 7a-b, and 8c).

2. Many studies throughout the manuscript need quantitation and further statistical analysis. For example, the authors could quantitate the overlap among GFP fusions and organelle markers in Supplementary Figure 1 (and other figures), the protein levels normalized to appropriate housekeeping genes for the blots in Figure 2A and Supplementary Figure 1C, and so on.

We agree with reviewer's comment. Accordingly, we quantified and normalized all of the immunoblot bands to tubulin (**new Figs. 2a, 2c~e, 3d, 4f, 5c, 5g, 6e** and **new Suppl. Figs. 3a, 7c**). In addition, the fluorescence intensity per cell in selected fields of the IFA images was quantified using the Zen software (Carl Zeiss, <http://zeiss.com>; **new Supplementary Fig. 1b**). We have also included detailed information on the statistical analysis for all bar graphs in the Method section of the revised

manuscript.

3. It is a little disconcerting that the authors perform various studies in at least five different cell lines (HFF, HELA, HEK293s, THP1s, and U937s). While there are advantages to working in the different systems for different assays, it is not explained why particular studies were performed in particular systems. This should be addressed.

We thank the reviewer for catching this oversight. We originally used HFF and HeLa cells to identify the specific functions of US7 and US8, because those cell lines are very useful for HCMV infection and protein overexpression, respectively. However, HeLa cells were not suitable for use in the luciferase-reporter assay to measure INF- β promoter activity, because they are not amenable to producing high expression levels of at least five different plasmids, including TLR3, TLR4/MD-2, and luciferase-expressing plasmid DNAs. Therefore, we used TLR3-expressing or TLR4/MD2-expressing HEK293T cells, which are widely used in IFN-reporter assays because of their high transfection efficiency. In addition, to determine if the blocking of TLR3 and TLR4 signaling by US7 and US8 was cell-type specific, we observed the downregulation of TLR4 cell-surface expression and IFN- β protein secretion in US7-expressing or US8-expressing immune cells of the monocyte-macrophage lineage (U937 and THP-1), which not only actively triggered TLR3/4 signaling, but also suggested to be a latency and persistence for HCMV. Therefore, we have revised the text throughout the revised manuscript to reflect the reviewer's points.

4. Much of the analysis of TLR3 and TLR4 is done with tagged proteins. The authors should repeat some of these studies with the endogenous TLRs. For example, can the authors observe US7 and US8 mediated destruction of the endogenous TLR proteins (Figure 2A) or demonstrate the effect of Sec61b/Derlin1 on endogenous TLR proteins (Figure 3D, E)?

The reviewer raises an important point about potential artefactual effects of the overexpression system. We performed an extensive co-IP and western blot analysis with endogenous proteins using anti-TLR3 and anti-TLR4 antibodies. Consistent with the results obtained using the tagged system, the immunoblot data showed considerable reduction of endogenous TLR3 and TLR4 protein levels. Together with Figs. 2b, 5e, and 5h and Supplementary Fig. 3c, we have provided the new data in the new Figs 2a, 3a, 3d, 4a, and 6e.

Minor concerns:

1. Lines 107 and 367. I'd rephrase this to avoid ambiguity and state that HCMV diminishes TLR3 and TLR4 protein levels (not expression, which refers to gene expression).

We thank the reviewer for identifying this ambiguity. We amended the sentences in the revised manuscript.

2. Line 124. While it is explained later, I'd add to this sentence that the changes in gene expression were monitored using microarrays.

We agree with reviewer's comment and have changed the text of the revised manuscript accordingly.

3. *Supplementary Figure 1 seems like a grab-bag of several studies. It might be preferable to split this into a localization supplemental figure and a signaling supplemental figure. At a minimum, this figure title could be altered.*

We agree with reviewer's comment and have split the old Supplementary Fig. 1 and moved it to the new **Supplementary Figs. 2a, 3b, or 3c** in the revised manuscript.

4. *Are the mRNA levels for TLRs in Supplemental Fig. 1C from the same study using the overexpressed myc-tagged proteins in Fig 2A? If so, this needs need to be clarified in Supplemental Fig 1C. If not, then why not?*

We thank the reviewer for catching this discrepancy. Because both US7 and US8 decrease TLR3 and TLR4 protein levels, we wondered if it was due to reduction of their mRNA levels. Thus, we have performed this RT-PCR experiment using cells only expressed either HA-US7 or HA-US8. As previously noted, to exclude potential artefactual effects of the overexpression, we performed western blot analysis with endogenous proteins using anti-TLR3 and anti-TLR4 antibodies (**new Figs. 2a, 3a, 3d, 4a, and 6e**). To avoid any confusion, we have revised the text and figures accordingly.

5. *Line 177. I'd delete "maturation" since this was not studied.*

We are grateful to the reviewer for catching this error. We have deleted "maturation" in the revised manuscript.

6. *The effects of Sec61b and Derlin1 shRNA seem somewhat mixed. Are the observed differences statistically significant?*

We agree with the reviewer's point that the restoration of US7-mediated or US8-mediated TLR3 and TLR4 degradation in cells expressing Sec61 β or Derlin-1 shRNAs was not clear in the previous version of the manuscript. We repeated those experiments with endogenous TLR3 and TLR4 to achieve more convincing results. While the knockdown levels were not quite as good as expected, consistent with the previous results with Myc-tagged TLR3 and TLR4, the immunoblot data showed considerable restoration of endogenous TLR3 and TLR4 protein levels. We also quantified and normalized with the corresponding TLR3 or TLR4 bands and provided the new datasets in **new Fig. 3d** in the revised manuscript.

7. *As the authors note, TLR4 localization will be affected by stimulation with LPS for example. It would be worth re-examining the effect of US8 on TLR4 localization in stimulated cells.*

The reviewer raises an important point about the effect of US8 on TLR4 localization in LPS-stimulated cells. TLR4 mainly localized in the ER/Golgi in control cells, but it localized in endosomes in LPS-stimulated cells. In LPS-stimulated cells expressed GFP-US8, the TLR4 was distributed in the Golgi and in punctate structures in lysosomal compartments in the presence of chloroquine, similarly to patterns in control cells. We have provided these data in the **new Supplementary Fig. 4**.

8. Line 266. I'd rephrase "protein turnover rate" since the authors aren't measuring this. Maybe protein accumulation?

We agree with reviewer's comment and have changed the text accordingly.

9. The control data for US7dCT that demonstrate that the truncated protein is there at comparable levels to the US7 full length protein (Supplemental Figure 2) displays GFP tagged constructs but the data in Figure 5B and Sup Figure 3C uses HA-tagged constructs. The protein level control should be performed using the same protein and optimally the same sample (the authors do provide this control for the other panels in Fig. 5).

We thank the reviewer for catching this discrepancy. Therefore, we repeated this experiment with both HA- and GFP-tagged versions to exclude global folding problems as an issue with regard to the deletion or point mutants. We have included the additional data in the **new Supplementary Fig. 5** of the revised manuscript.

10. The change in TLR4 ubiquitination in Fig. 5D doesn't look very profound.

We agree with the reviewer that the TLR4 ubiquitination patterns were not clear in the old Fig. 5d. Therefore, we repeated the experiment to achieve more convincing data. We have provided more convincing gels to clarify the TLR4 ubiquitination in US7-expressing cells (**new Fig. 5d**).

11. In light of the authors' observation that US8dCT doesn't localize properly, I'd be more cautious in how they interpret the role of the C-terminal domain in the function of that protein.

The reviewer points out that there were not sufficient data to conclude that "the C-terminal US8 region is essential for blocking TLR signaling" in the previous version of the manuscript, because we only showed a failure of lysosomal localization of US8 Δ CT and US8-Y204/211A. As described in the previous version, unlike the full-length US8, US8-Y204/211A failed to suppress the degradation of TLR3, TLR4 or UNC93B1, suggesting that tyrosine motifs in the US8 C-terminal domain are not only important for the lysosomal targeting of US8, it also play a role in blocking TLR signaling (**new Supplementary Fig. 7b**). However, we agree with the reviewer's point, because we still have not provided further support for direct roles of the C-terminal region in US8 function. Thus, we have replaced the term "critical/essential" with "involved" throughout the revised manuscript.

12. Lines 524-531. Please provide further details about the microscopy. On what system and with what software were the images captured, how were they processed and analyzed. How was it decided/quantitated whether proteins co-localized or not.

We included detailed information about the microscopy in the Methods section of the revised manuscript.

Comments from the reviewers that required a response are in bold italics, with each reply appearing in normal font just below the comment. In the replies, new material is emphasized in bold.

Reviewer #2 (Remarks to the Author):

The manuscript entitled “HCMV-encoded US7 and US8 act as antagonists of innate immunity by distinctively targeting TLR signaling pathways” by Park et al. describes a potential function of the unique short genes US7 and US8 to down regulate the TLR3 and TLR4 proteins. In general, the study analyzed TLR3 and TLR4 protein levels in cell systems that over-express wild type and mutant versions of US7 and US8. In addition, viral variants lacking US7 and US8 as well as rescued variants that express US7 and US8 only were analyzed for TLR3 and TLR4 stability and induction of inflammatory factors in virus infected cells. The manuscript describes some novel findings with regards to the function of US7 and US8 to modulate factors important for the innate immune response. However, there are some major issues with the manuscript that include a lack of controls that determine the specificity of US7 and US8 function, inconsistencies among the data with regard to the impact US7 and US8 have on TLR3/4 levels, the experiments are based on over-expressed cell systems, and the mechanism of US7 and US8 function to modulate innate immune regulators during the CMV life cycle is not sufficiently supported by experiments. Some of the specific items are outlined below.

1) Figure 1 nicely demonstrates that the expression of US7 and US8 modulates some of the inflammatory factors upon addition of dsDNA, polyIC, and LPS. However, the rationale to use polyIC and LPS is not clear given that polyIC and LPS are unlikely PAMPS for TLRs during a virus infection. In addition, the authors should include controls with cells that express other US genes that do not impact the inflammatory response.

The reviewer has raised the issue that we need to provide the rationale for using poly(I:C) and LPS in our study, because those are unlikely PAMPS for TLRs during viral infection. We first used dsDNA to identify the effects of US7 and US8 on global gene expression. Based on the results of microarray analysis, because the expression of IFN-related genes was significantly reduced by US7 and US8, and those US proteins predominantly localize in the ER or lysosomes, we originally attempted to target all TLRs, including TLR2, TLR3, TLR4, TLR7, and TLR9, along with STING/MAVS. We found that the cytokine gene expression induced by those TLR ligands was considerably decreased in cells expressing US7 or US8 (**new Supplementary Fig. 2b**). We only showed the results for TLR3 and TLR4 in the previous version of the manuscript, because we tried to focus on the IFN pathway triggered by TLR3 and TLR4, which is well known to be specialized for viral infection {Many previous studies suggest that both TLR3 and TLR4 have an important role in stimulating protective innate immunity against viral infection, including HCMV [Kawai, T. & Akira, S. et al., *Nat. Immunol.*, 2010; Takeuchi, O. & Akira, S. et al., *Cell*, 2010; Tabeta, K. et al., *PNAS*, 2004; Hutchens, M. et al., *J. Immunol.*, 2008; Kurt-Jones, E. A. et al., *Nat. Immunol.*, 2000; Okumura, A. et al., *J. Virol.*, 2010; Oshiumi, H. et al., *Nat. Immunol.*, 2003; Georgel, P. et al., *Virology*, 2007; Olejnik et

al., PLoS pathogens, 2018; Shinya et al., J. Virol., 2011; Wang et al., PLoS pathogens, 2019; Qin et al., Int J Clin Exp Pathol., 2018}}.

To avoid confusion and sufficiently provide a clear rationale for the use of TLR3 and TLR4 in our study, we have included new data showing the effects of US7 and US8 on the cytokine production elicited by ligands for activating TLR2, TLR3, TLR4, TLR7, and TLR9 signaling (**new Supplementary Fig. 2b**). Additionally, we included new data showing that there was no difference in TLR3-mediated or TLR4-mediated inflammatory-gene expression in cells overexpressing other HCMV-encoded US proteins, US14 and US15 (**new Supplementary Fig. 2c**).

2) In Figure 2, the down regulation of TLR3 and TLR4 is not very convincing and varies among the different panels. These experiments would greatly benefit from inclusion of controls that include cells that over-express other US genes and analysis of other membrane proteins such as MHC class I or transfected CD4. In addition, the radiolabeled studies have different exposures among the panels. The mature form of TLR4 can be observed in all conditions, despite the conclusion of the text.

We agree with the reviewer's comments, thus we performed an extensive immunoblot and co-IP analysis with endogenous TLR3 and TLR4 proteins to achieve more convincing results. Consistent with the results obtained with the tagged proteins, the immunoblot data show considerable downregulation of endogenous TLR3 and TLR4 protein levels, but not of MHC class I molecules. Together with Figs. 2b, 2c, 5e, and 5h and Supplementary Fig. 3c, we have provided new endogenous datasets in the **new Figs. 2a, 3a, 3d, 4a, and 6e**.

In addition, the reviewer pointed out an important issue with different exposures among the panels in the radiolabeled study. To avoid any confusion, we provided an original image not separated among panels.

3) The experiments in Figure 3 proposing that Derlin1- and Sec61beta are involved in US7 and US8 function to down regulate TLR3/4 is not very convincing. The lack of these genes had minimal impact on the levels of TLRs. Also, the inclusion of analyzing other membrane proteins as controls is lacking.

We agree with the reviewer's point that the restoration of US7-mediated or US8-mediated TLR3 and TLR4 degradation in cells expressing Sec61 β or Derlin-1 shRNAs was not clear. Therefore, we repeated the experiment with endogenous TLR3 and TLR4 to achieve more convincing results. While the knockdown levels were not quite as good as expected, consistent with the previous results obtained with Myc-tagged TLR3 and TLR4, the immunoblot data showed considerable restoration of endogenous TLR3 and TLR4 protein levels. We also performed the immunoblot analysis with other membrane protein, MHC class I molecules, in samples from cells expressing US7 or US8, but we observed no significant effects of US7 or US8 on the levels of these proteins. We have included these data in the **new Fig. 3d**.

4) The rationale and results of Figure 4 that examines UNC93B1 as the mechanism of US8

function is over interpreted.

We agree with reviewer's comment and have changed the text throughout the revised manuscript.

5) As with previous Figures, Figure 5 analyzing the US7 and US8 mutants did not have a major impact on TLR3/4 levels. Also, the data was over interpreted and does not support the manuscript's conclusions.

We agree with reviewer's comments that there are not sufficient data to conclude that “the C-terminal US8 region is essential for blocking TLR signaling.” As the reviewer has pointed out, because we still have not provided further support for direct roles of the C-terminal region in US8 function, we have replaced the term "critical/essential" with "involved in" throughout the revised manuscript.

6) Figure 6 is an important experiment that would provide relevance to the function of US7 and US8. The data support that a CMV infection decreases the levels of over-expressed TLR3 and TLR4. Yet, it would be quite important to examine endogenous levels of TLR3 and TLR4 as well as the protein expression of US7 and US8. In addition, the experiments should include the analysis of a non-specific cellular protein such as transferrin receptor or exogenously expressed CD4. Another issue was that the levels of TLR3 and TLR4 are inconsistent with the levels of inflammatory markers following a CMV infection.

The reviewer raises an important point about potential artefactual effects of the overexpression system. We therefore performed an extensive reciprocal immunoblot analysis with endogenous TLR3 and TLR4 in HFF cells infected with wild-type HCMV or HCMVΔUS7-16+US7/US8. Consistent with the immunoblot results obtained with US7-transfected or US8-transfected cells, the new data show **i)** considerable reduction of endogenous TLR3 and TLR4 protein levels in wild-type HCMV or HCMVΔUS6-17+US7/US8 infection, but not in HCMVΔUS6-17 infection, and **ii)** no significant difference in calnexin protein levels between infections with wild-type and mutant viruses. The new data are shown in the **new Fig. 6e**.

Comments from the reviewers that required a response are in bold italics, with each reply appearing in normal font just below the comment. In the replies, new material is emphasized in bold.

Reviewer #3 (Remarks to the Author):

Human cytomegalovirus (HCMV) encodes a large number of genes that subvert host immune response. Within a cluster of immune evasion genes in the US region of the HCMV genome are two, US7 and US8, whose function has been unknown. Park and colleagues now report their discovery that these two genes bind to and degrade TLR3 and TLR4. They provide some evidence that US7 acts by promoting degradation of Sec61b and Derlin-1 and that US8 disrupts the TLR3-UNC9381 interaction and promotes proteosomal and lysosomal targeting and degradation. They show the cytoplasmic tails of each protein are necessary for their phenotypes and importantly, that these proteins function in the context of viral infections.

Some data concerning the details of the mechanisms are not entirely convincing, but overall the studies have been well done, are quite extensive, and support that main, new and interesting conclusions.

Major comments:

1. The authors start this work by demonstrating that US7 and US8 repress the response to transfected dsDNA. If it is true that these genes function is to block TLR3 and TLR4 (sensors of dsRNA and of LPS, respectively) why do they work to block dsDNA?

We first used dsDNA to identify the effects of US7 and US8 on global gene expression, because HCMV has a dsDNA genome. Because the expression of IFN-related genes was significantly reduced by US7 and US8, and those US proteins predominantly localize in the ER or lysosomes, we originally attempted to target all TLRs, including TLR2, TLR3, TLR4, TLR7, and TLR9, along with STING/MAVS. We found that the cytokine production induced by those TLR ligands was considerably decreased in cells expressing US7 or US8 (**new Supplementary Fig. 2b**). We only showed the results for TLR3 and TLR4 in the previous version of the manuscript, because we tried to focus on the IFN pathway triggered by TLR3 and TLR4, which is well known to be specialized for viral infection {Many previous studies suggest that both TLR3 and TLR4 have an important role in stimulating protective innate immunity against viral infection, including HCMV [Kawai, T. & Akira, S. et al., *Nat. Immunol.*, 2010; Takeuchi, O. & Akira, S. et al., *Cell*, 2010; Tabeta, K. et al., *PNAS*, 2004; Hutchens, M. et al., *J. Immunol.*, 2008; Kurt-Jones, E. A. et al., *Nat. Immunol.*, 2000; Okumura, A. et al., *J. Virol.*, 2010; Oshiumi, H. et al., *Nat. Immunol.*, 2003; Georgel, P. et al., *Virology*, 2007; Olejnik et al., *PLoS pathogens*, 2018; Shinya et al., *J. Virol.*, 2011; Wang et al., *PLoS pathogens*, 2019; Qin et al., *Int J Clin Exp Pathol.*, 2018]}.

To avoid confusion and provide a clear rationale for the use of TLR3 or TLR4 in our study, we have included new data showing the effects of US7 and US8 on the cytokine secretion elicited by ligands activating TLR2, TLR3, TLR4, TLR7, and TLR9 signaling.

2. Several figures lack appropriate controls. For example, in Fig 3A, there is no control for nonspecific pull-down of US7. The two control lanes did not have any US7 in the input so of course there will be none in the pull down. What is needed is a control in which HA-US7 is transfected with an myc-tagged irrelevant control protein. This same argument applies to Fig 4A and 4D. Also, it would help to state how much of the lysate was loaded relative to the amount used for the IPs.

We agree that the co-IP experiments lacked appropriate controls. Therefore, we repeated the co-IP experiments with endogenous TLR3 and TLR4 to achieve more convincing results. To do that, we performed co-IP analysis with anti-TLR3 and anti-TLR4 antibodies to pull down endogenous TLRs. The new experiments provide more convincing results to clarify the interaction of both TLRs with US7 and US8 (**new Figs. 3a and 4a**). We also repeated the additional co-IP analysis with appropriate controls to show a more convincing result that US8 binds to UNC93B1 (**new Fig. 4d**). We added detailed information about the co-IP analysis in the Methods section of the revised manuscript.

3. Line 204, Fig. 3C. Derlin-1 appears to be pulled down by anti-HA even in the cell without either US11 or US7, undermining the interpretation that Derlin-1 associates with US7.

We previously performed the co-IP analysis with an anti-Derlin-1 antibody to pull down endogenous Derlin-1. The subsequent immunoblot assay was performed with an anti-HA antibody for US7. Thus, we were able to detect endogenous Derlin-1 in all lanes from the lysates or pull-down samples (in the right bottom panels of Fig. 3c). To avoid confusion, we highlighted the terms "IP: α -HA" and "IP- α -Derlin-1" in bold for emphasis in the **new Fig. 3c**.

4. Fig 3, D and E. How was US7 introduced into these cells? The impact of knocking down Derlin-1 on TLR4 stabilization is not convincing – 33.9 vs 28.3 is not a reliable difference in immunoblot assays. This result further weakens the argument that US7 acts through Derlin-1.

We agree with the reviewer's point that the restoration of US7-mediated or US8-mediated TLR3 and TLR4 degradation in cells expressing Sec61b or Derlin-1 shRNAs was not clear. We therefore repeated the experiment with endogenous TLR3 and TLR4 to achieve more convincing results. While the knockdown levels were not quite as good as expected, consistent with the previous results obtained with Myc-tagged TLR3 and TLR4, the new immunoblot data showed considerable restoration of endogenous TLR3 and TLR4 protein levels. We have included the new data in the **new Fig. 3d**.

5. Fig. S5. These results are hard to understand. Transfection of HFF is notoriously inefficient, so only a small fraction of cells is likely to be expressing the US7-10 genes. Therefore, the majority of cells that did not express the genes and were infected with the US7-16 genes would be expected to produce a large amount of interferon. Similarly, panel B results seem implausible. How do the authors account for these considerations?

Although HFF cells are not good for transfection, we tried to optimize the experimental conditions to achieve maximal transfection efficiency (>65%; please see the **additional Figure A** at the end of this

response). We performed a western blot analysis to check the expression levels of transfected HA-US7 and HA-US8 in HFF cells (Supplementary Fig. 8c, first and second panels). We also performed an RT-PCR experiment to show HA-tagged US7 and US8 overexpression in HFF cells. In HA-tagged US7-expressing or US8-expressing HFF cells, US7 and US8 mRNA expression levels were considerably higher than those in empty vector-expressing cells (**new Supplementary Fig. 8a**; please see the **additional Figure B** at the end of this response). Additionally, we provided detailed information about the HFF transfection in the Methods section of the revised manuscript.

Minor comments.

1. Line 786. “either 20 μ M MG132 for 4 hours” or what?

We thank the reviewer for identifying this error. We corrected the error in the revised manuscript.

2. The data in Fig. 4E do not show very convincing evidence for co-localization of UNC93B1 with US8, contrary to the text in line 244-5.

We agree with the reviewer that co-localization patterns of UNC93B1 with US8 were not clear. Therefore, we provided more convincing, sharper images to clarify the result in the revised manuscript (**new Fig. 4e**).

3. Line 266-268. The data in Fig S2A and S2B show quite a marked difference in the amount of the wt vs the deltaCT US7 and US8 under non-reducing conditions. How do the authors interpret

this observation?

We thank the reviewer for catching this discrepancy. Therefore, we repeated this experiment with both HA- and GFP-tagged versions to exclude global folding problems as an issue with regard to the deletion or point mutants. We have included the additional data in the **new Supplementary Fig. 5** of the revised manuscript.

4. Supplemental fig 3. The legend refers to transfection with empty vector in panel A, but no images are shown.

We thank the reviewer for catching this error. We included the additional image in the revised manuscript (**new Supplementary Fig. 6a**).

5. Line 310-311. It is not clear why inactivating the DNA sensor antagonist US9 was necessary.

We agree with the reviewer's comments and have amended the sentence in the revised manuscript.

6. The source or construction of the HCMV recombinants depicted in Fig. 6A should be described in the methods.

We thank the reviewer for identifying this lack of information. We have added detailed information regarding the source and construction of the HCMV recombinants in the Methods section.

Reviewers' Comments:

Reviewer #1:

Remarks to the Author:

The authors have successfully addressed my concerns, and the manuscript is now ready for publication in Nature Communications.

Reviewer #2:

Remarks to the Author:

The revised manuscript entitled "HCMV-encoded US7 and US8 act as antagonists of innate immunity by distinctively targeting TLR signaling pathways" by Park describes an interesting function of US7 and US8 to limit the innate immune response during a CMV infection. The authors addressed most of the concerns and the overall premise of the manuscript continues to be consistent throughout the paper. However, there are some technical points and conceptual points that need to be addressed.

- 1) The virus appears to modulate TL3 and TL4 activity following infection. Since these genes are involved in innate immunity, it is not clear why affecting the function of these TLR molecules would impact virus replication because when would these molecules encounter viral PAMPs during virus proliferation. This is fact is observed with the delta US7-US16 virus that is not important for replication and has no impact in viral replication. This point should be discussed in the paper.
- 2) The authors should revise their conclusion that TL3 and TL4 are down regulated from the cell surface because the data is histograms are not very convincing.
- 3) The authors need to clarify the IP conditions of Figure 3c because typically only mild lysis conditions allow recover of Sec61 associated proteins. Also, a control HA protein should be included in the experiment.
- 4) Figure 3e: anti-myc IP/western for TLR3 is not convincing in that there are no polypeptide bands.
- 5) Figure 3e: US7 expression is not convincing and looks like a background band.
- 6) Figure 4c: The enhanced ubiquitinylation data is not convincing and all samples in that experiment seem to have higher background exposure.

Comments from the reviewers that required a response are in bold italics, with each reply appearing in normal font just below the comment. In the replies, new material is emphasized in bold.

Reviewers' comments:

Reviewer #1 (Remarks to the Author):

The authors have successfully addressed my concerns, and the manuscript is now ready for publication in Nature Communications.

Reviewer #2 (Remarks to the Author):

The revised manuscript entitled “HCMV-encoded US7 and US8 act as antagonists of innate immunity by distinctively targeting TLR signaling pathways” by Park describes an interesting function of US7 and US8 to limit the innate immune response during a CMV infection. The authors addressed most of the concerns and the overall premise of the manuscript continues to be consistent throughout the paper. However, there are some technical points and conceptual points that need to be addressed.

1) The virus appears to modulate TL3 and TL4 activity following infection. Since these genes are involved in innate immunity, it is not clear why affecting the function of these TLR molecules would impact virus replication because when would these molecules encounter viral PAMPs during virus proliferation. This is fact is observed with the delta US7-US16 virus that is not important for replication and has no impact in viral replication. This point should be discussed in the paper.

We thank the reviewer for pointing out that we need to provide a rationale for concluding that US7-mediated or US8-mediated TLR3/4 downregulation impacts HCMV replication, because the Δ US7-16 mutation did not affect the replication of the virus. Previous studies suggest that HCMV encodes several different viral genes; such as UL44, UL82, UL83, UL122, or UL123, along with US7 and US8; which are all essential for the inhibition of IFN production and subsequent evasion of host antiviral responses (*Fu et al., J. Virol., 2019; Fu et al., Cell Host & Microbe, 2017; Biolatti et al., J. Virol., 2018; Abate et al., J. Virol. 2004; Kim et al., Front. Microbiol. 2017; Amsler et al., J. Mol. Biol. 2013; Marshall et al., J. Interferon & cytokine research, 2009*). That suggests the possibility of redundancy among those genes that might obscure important functions of the individual antagonists in the suppression of IFN responses under different conditions. For example, HCMV produces four different G protein-coupled receptors (GPCRs): US27, US28, UL33, and UL78 (*Ahuja et al., J. Biol. Chem., 1993; Cesarman et al., J. Virol. 2004; Isegawa et al., J. Virol. 1998; Margulies et al. Virology, 1996; Waldhoer et al., J. Virol. 2002; Milne et al., J. Immunol. 2000; Neote et al., Cell, 1993, Gao et al., J. Biol. Chem., 1994*). Deletion of all four GPCRs is required to produce a complete defect in replication in both fibroblasts and endothelial cells (*Miller et al., PLoS One, 2012; O'Connor et al., J. Virol. 2010, 2012*). In a similar context, the HCMV Δ US7-16 mutant virus might use other viral proteins, such as UL44, UL82, UL83, UL122, or UL123, which serve rescue the

blockade of the IFN response in place of US7 and US8, thus maintaining viral replication in the host. We have revised the Discussion section of the manuscript to reflect those important considerations.

2) The authors should revise their conclusion that TL3 and TL4 are down regulated from the cell surface because the data is histograms are not very convincing.

To provide more convincing results, we repeated FACS experiments and presented the results with dot plots as well as histograms to show the cell-surface expression level of TLR4 (we only analyzed TLR4 surface expression levels, because TLR3 proteins are ultimately transported to the endolysosomes, where they encounter and respond to their ligands).

Consistent with the results obtained with the immunoblot analysis, showing significant downregulation of endogenous TLR3 and TLR4, both the dot plots and the histograms from the FACS analysis showed considerable downregulation of TLR4 cell-surface expression in both US7-expressing and US8-expressing cells (**new Figure 2b**). In addition, the cell-surface expression of TLR4 was restored in cells expressing US7 Δ CT, US8 Δ CT, or US8-Y204/211A, to the extent of being essentially indistinguishable from that in empty-vector-expressing cells (**new Figure 5e and 5h**). Additionally, TLR4 cell surface expression levels from all FACS data were evaluated with the statistical analysis. We thus provided the new FACS datasets (dot plots), as well as the statistical graphs in the **new Figures 2b, 5e, and 5h**. Please see the histograms and dot plots at the end of this response.

3) *The authors need to clarify the IP conditions of Figure 3c because typically only mild lysis conditions allow recover of Sec61 associated proteins. Also, a control HA protein should be included in the experiment.*

We thank the reviewer for identifying this lack of information. For performing the co-IP experiments for Sec61 β -US2/US11/US7 interactions, cells were lysed with 1% digitonin with protease inhibitors and the lysates were then incubated with anti-HA (**Fig. 3c**) or anti-GFP antibody (**new Supplementary Fig. 3d**) for 6 h followed by protein G-Sepharose beads for 1 h at 4°C. The beads were then washed three times with 0.1% digitonin and the eluted proteins were eluted by boiling the samples in denaturing buffer.

As the reviewer pointed out, we only used the US2 protein capable of binding Sec61 β as a positive control in the previous version of the manuscript, because it was difficult to determine which HA-tagged proteins would be appropriate negative controls. Therefore, to confirm our results with a negative control, we performed an additional co-IP analysis with GFP-tagged US7, using GFP alone as a negative control. We observed a clear co-IP band indicating that Sec61 β indeed interacts with US7-GFP, but not with GFP alone (**new Supplementary Fig. 3d**; please see the co-IP result at the end of this response).

We have thus added detailed information regarding the co-IP conditions for the Sec61 β /Derlin-1-US2/US11/US7 interactions or for TLR3/4-US7/8 or UNC93B-US8 interactions in the Methods section and Supplementary Figure legends.

4) *Figure 3e: anti-myc IP/western for TLR3 is not convincing in that there are no polypeptide bands.*

We agree with the reviewer. To achieve more convincing results with ubiquitination assay, we used HEK293T cells (instead of HeLa cells) that expressed TLR3-Myc or TLR4-Myc to obtain maximal expression efficiency and to determine the clear TLR3/TLR4 yield after immunoprecipitation with anti-Myc antibody. Additionally, we performed this ubiquitination analysis with cells treated with 20

μ M MG132 for 4 h to prevent US7-mediated degradation to obtain sufficient amounts of TLR3 or TLR4 proteins. We have thus provided more convincing images to clarify the US7-mediated TLR3 and TLR4 ubiquitination. The new datasets are provided in the revised manuscript (**new Figure 3e**).

5) Figure 3e: US7 expression is not convincing and looks like a background band.

We agree with the reviewer and repeated this experiment with HEK293T cells to achieve more convincing results and provided new datasets in the revised manuscript.

6) Figure 4c: The enhanced ubiquitinylation data is not convincing and all samples in that experiment seem to have higher background exposure.

We agree with the reviewer that the TLR4 ubiquitination patterns were not clear in the old **Figure 4c**. Therefore, we repeated the experiment with HEK293T cells to obtain both maximal expression efficiency and greater yields of TLR4 proteins after immunoprecipitation. The additional experiments produced more convincing gels to clarify the TLR4 ubiquitination in US8-expressing cells (**new Figure 4c**).

Reviewers' Comments:

Reviewer #2:

Remarks to the Author:

The revised manuscript has adequately addressed my concerns and I recommend publication.